# Chiral terahertz wave emission from the Weyl semimetal TaAs

Y. Gao [1], S. Kaushik[2], E.J. Philip [2], Z. Li[3,4], Y. Qin[1,5], Y.P. Liu[6], W.L. Zhang[1], Y.L. Su[1], X. Chen[2], H. Weng [4,7], D.E. Kharzeev [2,8,9*], M.K. Liu[2*] & J. Qi [1*]

Weyl semimetals host chiral fermions with distinct chiralities and spin textures. Optical excitations involving those chiral fermions can induce exotic carrier responses, and in turn lead to novel optical phenomena. Here, we discover strong coherent terahertz emission from Weyl semimetal TaAs, which is demonstrated as a unique broadband source of the chiral terahertz wave. The polarization control of the THz emission is achieved by tuning photoexcitation of ultrafast photocurrents via the photogalvanic effect. In the near-infrared regime, the photon-energy dependent nonthermal current due to the predominant circular photogalvanic effect can be attributed to the radical change of the band velocities when the chiral Weyl fermions are excited during selective optical transitions between the tilted anisotropic Weyl cones and the massive bulk bands. Our findings provide a design concept for creating chiral photon sources using quantum materials and open up new opportunities for developing ultrafast opto-electronics using Weyl physics.

[1] State Key Laboratory of Electronic Thin Films and Integrated Devices, University of Electronic Science and Technology of China, Chengdu 611731, China. [2] Department of Physics and Astronomy, Stony Brook University, Stony Brook, NY 11794, USA. [3] State Key Laboratory for Artificial Microstructure and Mesoscopic Physics, Beijing Key Laboratory of Quantum Devices, Peking University, Beijing 100871, China. [4] Beijing National Laboratory for Condensed Matter Physics, Institute of Physics, Chinese Academy of Sciences, Beijing 100190, China. [5] Institute of Electronic and Information Engineering, University of Electronic Science and Technology of China, Dongguan 523808, China. [6] Institute of Modern Physics, Fudan University, Shanghai 200433, China. [7] Songshan Lake Materials Laboratory, Dongguan 523808, China. [8] Department of Physics, Brookhaven National Laboratory, Upton, NY 11973-5000, USA. [9] RIKEN-BNL Research Center, Brookhaven National Laboratory, Upton, NY 11973-5000, USA. *email: dmitri.kharzeev@stonybrook.edu; mengkun.liu@stonybrook.edu; jbqi@uestc.edu.cn

The generation and control of photoinduced charge current and the resultant electromagnetic wave emission are of crucial importance for coherent operation in opto-electronic quantum devices[1]. The merit is twofold. First, the photocurrents induced by optical transitions obeying the selection rules and/or chirality of the materials naturally permit ultrafast manipulation. This is especially true when the nonthermal excitation of both charge and spin degrees of freedom can be utilized[2,3]. Second, the emission of electromagnetic wave induced by ultrafast currents is essential in the terahertz (THz) frequency regime, where control of the ellipticity and chirality over a broad spectral range is known to be notoriously difficult[4–9]. Specifically, previous generation and manipulation of the broadband helical THz wave mainly rely on sophisticated pulse shaping or two color/pulse manipulation of the incident light[4–9], where the emitter itself permits no intrinsic optical chiralites. For example, conventional nonlinear crystals, such as ZnTe and $LiNbO_3$, only allow THz emission linearly polarized along one particular crystal axis, irrespective of the polarization of the incident light. Therefore, in order to achieve the polarization control over a wide spectral range at the THz regime, one promising novel way is to exploit the topology of band structures where the carriers demonstrate unique spin-momentum locking or chiral properties[10–12].

Weyl semimetals (WSMs), with their intriguing topological electronic structure, retain chiral electrons near the Weyl nodes[13–19]. Previous work has proposed that the emergence of various fascinating electronic responses to light is intimately associated with the Berry phase of the topological bands, e.g., spin-polarized photovoltaic currents[20–22], photoinduced anomalous Hall effect[23], and quantized photocurrents[24]. Therefore, the investigation of photocurrents in WSMs has raised enormous interest both theoretically and experimentally[20–28]. For mid-infrared light, refs. [20,21,25] show the existence of a dominant helicity-dependent DC photocurrent owing to the circular pho-togalvanic effect (CPGE) in the WSM TaAs. In contrast, with linearly polarized light, a giant linear photogalvanic effect (LPGE) (or shift current) was observed in ref. [26]. At the near-infrared regime, photocurrent measurements[27] suggest the existence of CPGE in TaAs. The THz wave emission from WSM TaAs using ~1.5 eV photon was also reported recently[29], which is interpreted to arise predominantly from the CPGE-induced photocurrents. Similar THz emission was also observed in the ferroelectric semiconductors[30,31], where the LPGE across the bandgap in a wide photon-energy range has been studied. However, up to now for WSMs upon excitation of high-photon energies with an order of ~1 eV, it is still quite elusive whether the Weyl physics has an essential role in the giant nonlinear optical responses including the second harmonic generation[32,33] and the photocurrents. Although analysis of the crystal symmetry can reveal all the possible components of the nonthermal photocurrent[26,28,29], it fails to provide information about the magnitude or photon-energy dependence of the current. Therefore, a theoretical study of the underlying mechanism along with an experimental inves-tigation across the entire near- and mid-infrared ranges are mandatory.

In this paper, we reveal the generation of chiral (or helical) broadband THz waves in the WSM TaAs. We find that the polarization of these THz waves can be easily manipulated without incorporating any THz waveplate. Such polarization control arises from the colossal ultrafast photocurrents whose direction and magnitude depend on the polarization (circularly or linearly polarized) of the femtosecond optical pulses. For the first time, the photon-energy-dependent ultrafast photocurrents in TaAs have been quantitatively determined at the near- and mid-infrared light frequencies. In addition, a careful theoretical

treatment suggests that the chiral Weyl fermions indeed play a crucial role in the generation of the ultrafast photocurrents owing to the dominant CPGE at high-photon energies.

## Results

**THz emission from TaAs**. A schematic of the experiment is shown in Fig. 1a. Femtosecond laser pulses are used to induce ultrafast photocurrents. According to the Maxwell equations, a change in the current density $\mathbf{j}(z, t)$ on the picosecond (ps) timescale will result in electromagnetic radiation in the THz spectral range (1 THz = 1 $ps^{-1}$)[34]. The transient electric field $\mathbf{E}(t)$ is generated with a polarization parallel to the direction of the current. Therefore, one can use the time-domain spectra $\mathbf{E}(t)$ of the THz radiation as a probe for the ultrafast sheet current density given by $\mathbf{J}(t) = \int \mathbf{j}(z, t) dz$. The orthogonal components $J_x$ and $J_{yz}$ via the generalized Ohm's law determine the THz near-field $\mathbf{E}(t)$ on top of the sample surface, i.e., the s-polarized $E_x$ along $\hat{\mathbf{x}}$ and the p-polarized $E_{yz}$ in the $\hat{\mathbf{y}}\hat{\mathbf{z}}$ plane. Experimentally, the THz far-field electro-optic (EO) signal $\mathbf{S}(t)$ was collected, and the THz near-field $\mathbf{E}(t)$ was derived via inversion procedures based on a linear relationship between $S(\Omega)$ and $E(\Omega)$[34] (see Methods). Therefore, $\mathbf{S}(t)$ is a qualitative indicator of the ultrafast photocurrent, whose genuine properties shall be quantitatively obtained by analyzing $\mathbf{J}(t)$.

In the present experiments, unless noted in the text, we mainly focus on the results obtained for the TaAs(112) single crystal with an incident angle $\Theta \simeq 3°$ using the excitation light with a wavelength of 800 nm. $\hat{\mathbf{x}}$ is along the [$\bar{1}$10] direction. The results for (011) and (001) faces, another angle of incidence ($\Theta \simeq 45°$), and other excitation wavelengths are reported in the Supplemen-tary Figs. 6–12. Figures 1b–d show strong time-domain THz far-field EO signals $\mathbf{S}(t)$ detected from the sample. Within our pump power range, the THz peak field strength can reach up to ~1 kV $cm^{-1}$ and the dynamic range of $\mathbf{S}(t)$ can be ~60 dB (see Supplementary Fig. 5). Clearly, both the magnitude and temporal shape of the THz waveform $S_x(t)$ depend strongly on the light polarization. The key observation is that signals $S_x(t)$ taken with right- (↻) and left-handed (↺) circularly polarized light are completely out of phase (Fig. 1b). A similar observation was found for the 45° and 135° linearly polarized light (Fig. 1c). In terms of the peak values, $S_x$ induced by the linearly polarized light is approximately four times smaller than that owing to excitation by circularly polarized light. By contrast, $S_{yz}(t)$ is almost polarization-independent and differs substantially from $S_x(t)$ (see Fig. 1d). Such distinct $S_x$ and $S_{yz}$ components lead to an elliptically polarized transient THz field $\mathbf{S}(t)$ (or $\mathbf{E}(t)$), which exhibits opposite chirality for different circularly or linearly polarized pump light (see Fig. 2a and b), as we will discuss in detail below.

In the frequency domain of the THz near-field $\mathbf{E}(t)$ (see Fig. 3a–b), the dominant spectra for both $E_x$ and $E_{yz}$ sit below ~3 THz. At ~1.7 and 3.1 THz (~57 and 104 $cm^{-1}$), there exist two obvious dips. The former might be owing to the infrared active phonon mode in TaAs. The latter can be attributed to the absorption of the Raman active $E(1)$ mode[35]. The high-frequency tails extend almost to 12 THz, consistent with a time resolution of ~80 fs. Strikingly, for $0.2 \lesssim \Omega \lesssim 3$ THz, we discovered that the phase difference, $\Delta\varphi$, between $E_x$ and $E_{yz}$ is nearly constant. This phase difference is independent of the incident angle $\Theta$ for a given pump polarization. However, its value differs between different faces (insets of Fig. 3a and b), i.e., for (112), $\Delta\varphi_{↺} \simeq \pi/3$ and $\Delta\varphi_{↻} \simeq 4\pi/3$; for (011), $\Delta\varphi_{↺} \simeq \pi/2$ and $\Delta\varphi_{↻} \simeq 3\pi/2$. The origin of $\Delta\varphi$ will be discussed later. Nevertheless, such extraordinary findings suggest that $\mathbf{E}(t)$ can be well regarded as a broadband elliptically polarized THz pulse with its detailed characteristics

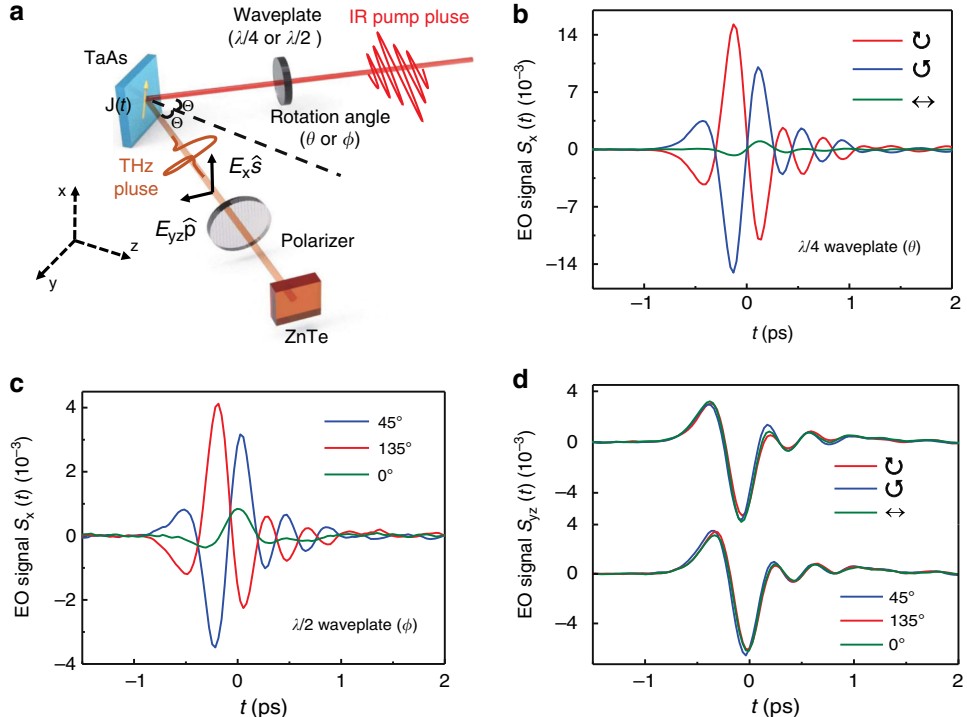

**Fig. 1 THz emission from the Weyl semimetal TaAs. a** Schematic of the THz emission spectroscopy. Excitation of a fs laser pulse with an incident angle $\Theta$ onto a TaAs single crystal initiates a photocurrent burst and, consequently, emission of a THz pulse $\mathbf{E}(t)[= E_x(t)\hat{s} + E_{yz}(t)\hat{p}]$. Measurement of the components $E_x(t)$ and $E_{yz}(t)$ by the THz EO sampling provides access to the sheet current density $\mathbf{J}(t)$ flowing inside the sample. **b–d** Typical THz EO signal components $S_x(t)$ and $S_{yz}(t)$ along the $\hat{s}$ and $\hat{p}$ directions were measured at various settings for pump polarization via rotating the $\lambda/4$ or $\lambda/2$ waveplate, characterized by the angle $\theta$ or $\phi$, respectively. Here, $\leftrightarrow$ ($\theta = 0°$), ↻ ($\theta = 45°$), and ↺ ($\theta = 135°$) represent the p, right-handed, and left-handed circularly polarized light, respectively. The angle $\phi$ stands for the linear polarization state with respect to the p-polarized light ($\phi = 0°$).

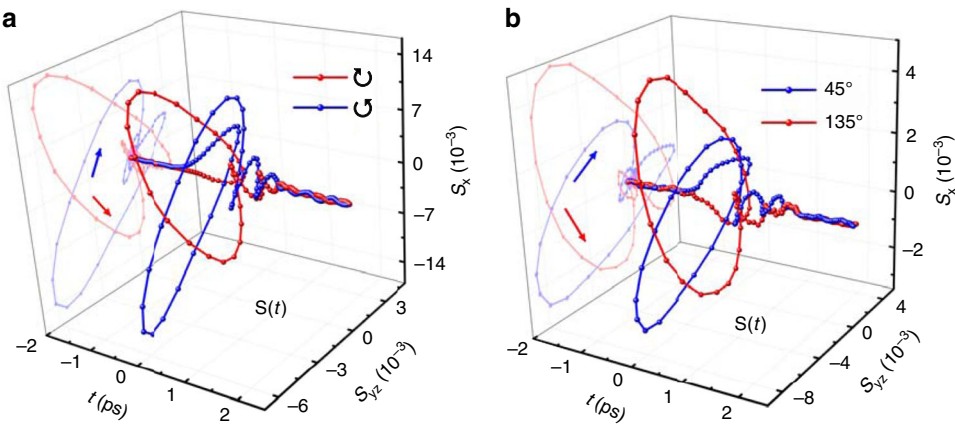

**Fig. 2 The far-field EO signals $S(t)[= S_x(t)\hat{s} + S_{yz}(t)\hat{p}]$ from TaAs(112). a** and **b** are for circularly and linearly polarized pump light, respectively. The colored arrows indicate different optical chiralities of the emitted THz wave: left-handed (blue) and right-handed (red).

depending on the pump polarization and the sample faces and, hence, has a defined chirality. According to the polarization trajectory ($S_{yz}(t)$, $S_x(t)$) in Fig. 2a and b, chirality of the helical THz pulse can be instantaneously switched by varying the circular or linear polarization of the pump light.

**Polarization dependence of the THz signals**. Next, we elucidate the mechanism underlying the THz emission. We measured the dependence of $S_x(t)$ and $S_{yz}(t)$ on the degree of circular polarization of the incident light, which can be controlled by rotating the quarter-waveplate by an angle $\theta$ (Fig. 1). The experimental

results were found to be well fitted by the following equation[26,36,37]

$$S_\lambda(t, \theta) = C_\lambda(t) \sin 2\theta + L_{1\lambda}(t) \sin 4\theta + L_{2\lambda}(t) \cos 4\theta + D_\lambda(t),$$

(1)

where $\lambda = x$ or $yz$. $C_\lambda$ represents the contribution from helicity-dependent photocurrents. $L_{1\lambda}$ depends on the linear polarization and is phenomenologically associated with a quadratic nonlinear optical effect. $L_{2\lambda}$ and $D_\lambda$ arise from a thermal effect related to the light absorption. All four terms on the right side of Eq. (1)

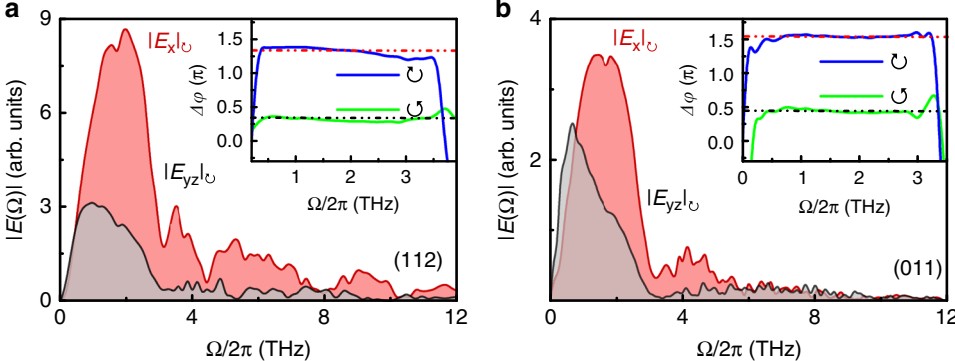

**Fig. 3 Fourier transformed spectra for the THz near-field E(t). a** and **b** are from (112) and (011) faces for circularly polarized pump light, respectively. The insets show the phase difference ($\Delta\varphi$) between $E_x(\Omega)$ and $E_{yz}(\Omega)$ for each circularly polarized pump light. The dashed lines represent the average value of $\Delta\varphi$.

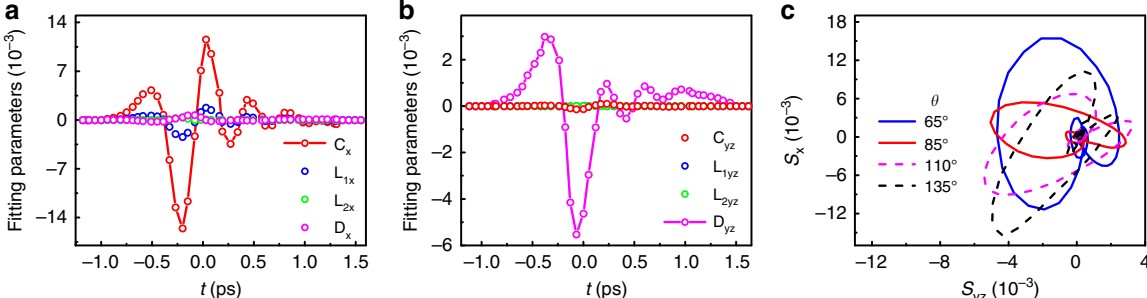

**Fig. 4 The obtained THz EO signals by rotating the quarter-wave plate. a–b** Time-dependent fitting parameters $C_\lambda$, $L_{1\lambda}$, $L_{2\lambda}$, and $D_\lambda$ ($\lambda = x$, $yz$) in Eq. (1). **c** shows the polarization trajectory ($S_{yz}(t)$, $S_x(t)$) under elliptically polarized pump light with different $\theta$. The solid and dashed curves represent opposite chiralities.

depend monotonically on the optical pump power, which agrees with our experimental observation (see Supplementary Fig. 5).

Figures 4a and b display the time-dependent parameters $C_\lambda$, $L_{1\lambda}$, $L_{2\lambda}$, and $D_\lambda$, which were obtained by fitting the experimental $S(t, \theta)$ using Eq. (1). $S_x(t)$ and $S_{yz}(t)$ as a function of $\theta$ can be found in the Supplementary Fig. 5. Based on our results, $S_x$ is unambiguously dominated by $C_x$, and has a non-negligible contribution from $L_{1x}$. Both the amplitude and phase of $S_x(t)$ change with $\theta$, whereas $L_{2x}$ and $D_x$ can be omitted. On the other hand, $S_{yz}(t)$ is dominated by a polarization-independent $D_{yz}(t)$. $C_{yz}$ has a very small role, whereas $L_{1yz}$ and $L_{2yz}$ can be neglected. These results suggest that the ultrafast photocurrents leading to the THz signal $E_x$ (or $S_x$) is polarization-dependent (or $\theta$-dependent), in contrast to the polarization-independent thermally related photocurrent inducing $E_{yz}$ (or $S_{yz}$). Therefore, as demonstrated in Fig. 4c, one can control the ellipticity and chirality of the helical THz pulse by changing ellipticity of the pump light. Realization of circularly polarized THz pulses also becomes possible, e.g., THz emission from the (011) face with $\Delta\varphi \simeq \pi/2$ (see Supplementary Fig. 7).

Photocurrents arising from the linearly polarized light can be uncovered by measuring the dependence of $S_x$ and $S_{yz}$ on the linear polarization angle $\phi$, using a half-wave plate. The angle dependence of $S_x(t, \phi)$ near the peak values are shown in Fig. 5a and b. We found that $S_x(\phi)$ can be well described by a second-order nonlinear optical process after considering the crystal symmetry, as demonstrated by the fitted curves in Fig. 5a and b (see Methods and Supplementary Note 1 for detail fitting equations). On the other hand, $S_{yz}$ is only slightly modulated by the linear polarization-dependent signal and is dominated by a polarization-independent background. Similarly,

we can manipulate the chiral THz pulse by changing the linear polarization state of the pump light, as illustrated in Fig. 5c.

**Ultrafast photocurrents in TaAs.** Observations of chiral broadband THz pulses indicate that the amplitude and phase of the ultrafast photocurrents can be fully controlled by polarized fs optical pulses. One can use the measured THz signals to quantitatively extract the ultrafast photocurrents (see Methods), which are displayed in Fig. 6a and b. The data unambiguously demonstrate that switching of the current direction of $J_x(t)$ occurs instantaneously using circularly or linearly polarized light, whereas $J_{yz}(t)$ is nearly unchanged for different polarized light. Along the time axis, **J**(t) shows helical behavior. As a result, **J**(t) has chirality, which can be manipulated by the polarized pump light. With regard to the dynamics of $J_x(t)$ and $J_{yz}(t)$, after an initial onset, $J_x(t)$ generally proceeds much faster than $J_{yz}(t)$. The former notably shows a strong oscillatory behavior, which might be attributed to plasma oscillation (or plasmon) of the charge carriers. In fact, both previous Fourier transform infrared spectroscopy[38] and our ultrafast optical transient reflectivity studies (see Supplementary Fig. 13) show that the Drude scattering time in TaAs has a timescale of ~400 fs, which is consistent with the current relaxation time, defined by the time required for the peak current density reducing by a factor of $1/e$ (see Fig. 6a and b).

## Discussion

To determine the origin of $J_x$ and $J_{yz}$, we need to consider the mechanisms for the photocurrent generation. Analysis based on group theoretical arguments alone can shed light on all the possible components that can contribute to photocurrent, but

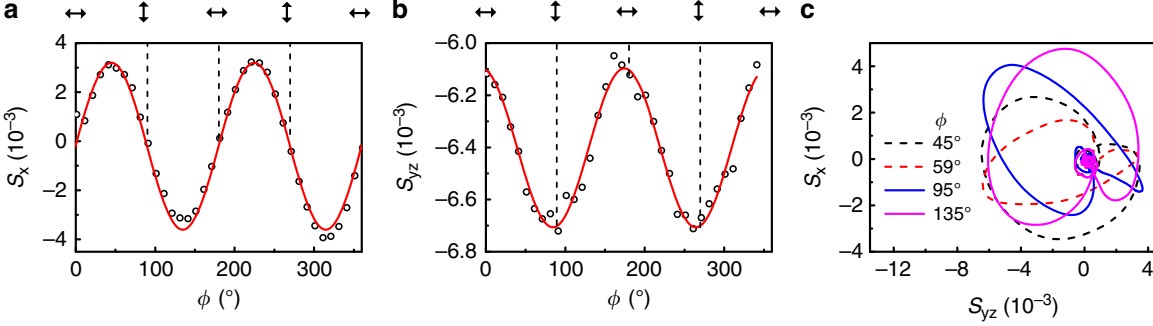

**Fig. 5 The obtained THz EO signals by rotating the half-wave plate. a** and **b** display $S_x(t = 0.03 \text{ ps})$ and $S_{yz}(t = -0.02 \text{ ps})$ (near the peak values) as a function of the linear polarization angle, $\phi$, respectively. The red solid lines show the fitted results. **c** shows the polarization trajectory $(S_{yz}(t), S_x(t))$ for different linearly polarized pump light with several typical $\phi$. The solid and dashed curves represent opposite chiralities.

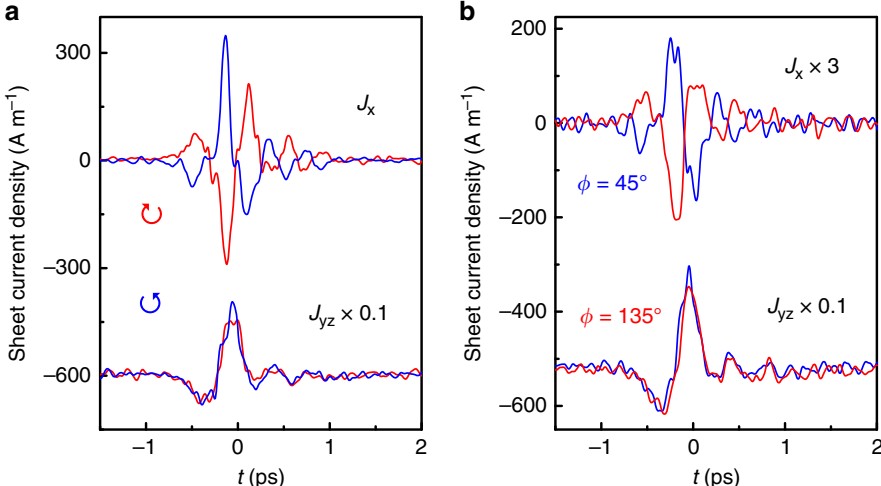

**Fig. 6 Extracted sheet photocurrent densities $J_x(t)$ and $J_{yz}(t)$. a** and **b** are for different circular and linear pump polarization at 800 nm, respectively. Curves are offset for clarity.

will not offer any information regarding their magnitude or photon-energy dependence. Microscopically, the ultrafast photocurrents can be generated during processes such as optical transitions, phonon- or impurity scatterings, and electron-hole recombinations[39]. Photocurrents induced by the optical transitions, occurring within the pulse duration, can in principle be controlled non-thermally in an ultrafast way[20,25,36,37]. Of particular interest are the photocurrents owing to the CPGE and LPGE[26,36], which are often, respectively, referred to as injection and shift currents[40,41]. In the case of TaAs, based on the obtained sheet current densities, the injection currents owing to CPGE play the dominant role in $J_x(t)$.

We carried out detailed theoretical calculations to clarify our observations for CPGE. The schematics is displayed in Fig. 7a, where the circularly polarized light introduces asymmetric population/depopulation of the excited (initial or final) states following particular optical selection rules. For carrier excitations in between the Weyl bands using small photon energies, the Pauli blockade and cone tilting are vital for generating the helicity-dependent DC photocurrent in TaAs[20,25]. However, as much higher photo energies (>470 meV) are used in our experiment, excitation within the Weyl cone will not happen. Instead, interband transitions can occur between the massless Weyl cones and the massive bulk states far away from $E_f$[17,42]. In such case, the quasiparticles are excited from/to a linear Weyl band with a large

momentum-independent velocity $v = \partial E(q)/\partial q$ to/from a band with much smaller velocity, and lead to a large velocity difference between the initial and final states. This effect potentially will induce a large non-zero photocurrent. If this current relaxes over a timescale of ~1 ps, i.e., rapid deceleration of electric charges, the electromagnetic radiation in the THz frequency range can take place, as observed in our work.

We derive the expression for CPGE in the situation where optical transition occurs from a linear Weyl band to a massive band. The calculation would be analogous for transitions from a massive band to a linear Weyl band. This is because compared with the electrons, the holes have opposite helicity and charge, which result in the final expression having the same sign. Our calculations are restricted to the photon-energy range where optical transitions involve the linear Weyl band, as shown in Fig. 7a. At higher photon energies, the transitions will start to occur from the chiral band without linear dispersion to the massive band, which reduces the THz emission. At lower photon energies, the optical transitions between the Weyl cones and the massive bands would rarely happen owing to the lack of joint density of states[17,42]. In the latter two cases, the CPGE-induced photocurrent will drop quickly to zero. For photon energies down to ~120 meV, interband transitions within the Weyl cones will emerge, and the associated photocurrents have already been investigated intensively[20,21,24–26,28].

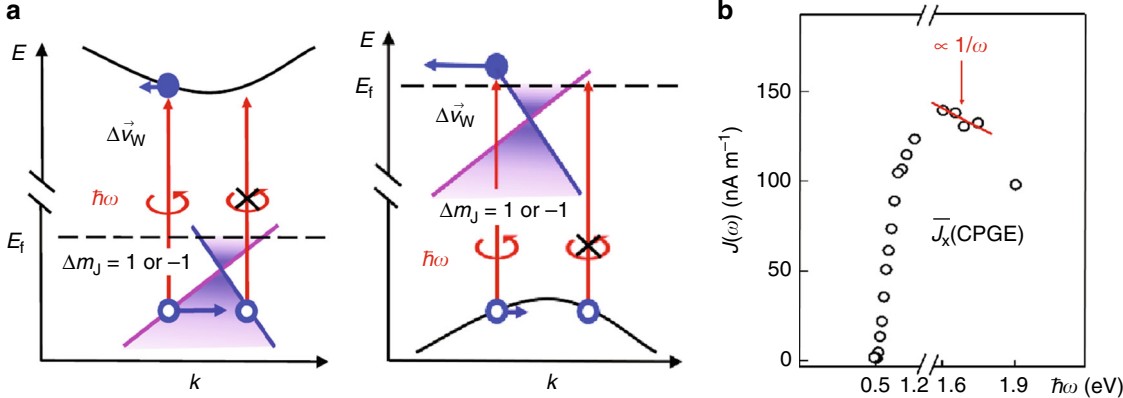

**Fig. 7 Schematic of CPGE and its dependence on the incident photon energy. a** The asymmetric population or depopulation of the tilted Weyl cones leading to an average band velocity of $\Delta\mathbf{v}_\mathrm{w}$ results in a non-zero charge current using circularly polarized light. The dashed line represents the Fermi level[25]. During the optical transitions, the total angular momentum should be conserved, and its quantum number $m_J$ satisfies the relation: $\Delta m_J = 1$ or $-1$, depending on the helicity of the pump pulse. The left is for excitations from the linear Weyl bands to the high-lying massive states. The right is for excitations from the low-lying massive bands to the linear Weyl bands. **b** The DC sheet current density $\overline{J}_\mathrm{x}(\omega)$ as a function of the photon energy, $\hbar\omega$. $\overline{J}_\mathrm{x}(\omega)$ is expected to be proportional to $1/\omega$ in the near-infrared spectral region according to our theoretical model, as indicated by the red solid line.

We assume the Hamiltonian in which a single Weyl cone makes a contribution is described by

$$\hat{H} = \hbar v_\mathrm{a}^\mathrm{i}\, \sigma_\mathrm{a}\, q_\mathrm{i} + \hbar v_\mathrm{t}^\mathrm{i}\, \sigma_0\, q_\mathrm{i} \equiv \hat{H}_\mathrm{W} + \hat{H}_\mathrm{t} + E_0, \quad (2)$$

where $\sigma_\mathrm{a}$ are the Pauli matrices, $\sigma_0$ the identity matrix, $a$ is the pseudospin index and $i$ is the spatial index; the vector $\mathbf{q}$ is the momentum measured from the Weyl point; $E_0$ is the energy of the Weyl point. The first term $\hat{H}_\mathrm{W}$ contains information about the chirality and the velocity of the Weyl fermion, and the second term $\hat{H}_\mathrm{t}$ describes the tilt in the direction determined by the constant vector $\mathbf{v}_\mathrm{t}$; $q_\mathrm{i}$ is the quasiparticle's momentum measured from the position of the Weyl node; $v_\mathrm{ia}$ is the velocity matrix. The interaction with the electromagnetic field $\mathbf{A}$ is obtained from 2 through the Peierls substitution $\mathbf{q} \rightarrow \mathbf{q} - \frac{e}{\hbar}\mathbf{A}$; this leads to the electromagnetic interaction Hamiltonian $\hat{H}_\mathrm{EM}$

$$\hat{H}_\mathrm{EM} = -e v_\mathrm{a}^\mathrm{i}\, \sigma_\mathrm{a}\, A_\mathrm{i} - e\, v_\mathrm{t}^\mathrm{i}\, \sigma_0\, A_\mathrm{i} \equiv \hat{H}_\mathrm{WEM} + \hat{H}_\mathrm{tEM}. \quad (3)$$

Here, the second term (which we denote by $\hat{H}_\mathrm{tEM}$) is diagonal in spin space and does not contribute if one considers transitions between the Weyl bands; however, it will in general contribute once other nonlinear bands are excited.

Using the Hamiltonian 3, the induced electric current density can now be readily computed based on two physical assumptions: (i) we can neglect the band velocity of an excited band compared with the velocity on the Weyl band[17,42], so the energy of the excited band can be assumed to be approximately independent of momentum (flat band approximation); and (ii) once the photons enter the material, they will induce an excitation with unit probability. Assumption (ii) may not be realistic owing to other excitations induced by the photons, e.g., the shift photocurrents discussed in later sections. Based on these assumptions and using Fermi's golden rule, we can write the current density integrated over the penetration depth (the DC sheet current density) as

$$
\begin{aligned}
\mathbf{J}(\omega, \mathbf{k}_\mathrm{p}, \boldsymbol{\varepsilon}) &= \int \mathbf{j}(\omega, \mathbf{k}_\mathrm{p}, \boldsymbol{\varepsilon})\, \mathrm{d}z \\
&= \frac{-eI}{\hbar\omega} \frac{\sum_l \tau_l a_l \int \frac{\mathrm{d}^3 q}{2\pi^3}\, \delta(E_{l-}(q) - E_{l1} + \hbar\omega)\, (0 - \mathbf{v}_{l-}(q))\, \sum_i |\langle s_{li}|\hat{H}_\mathrm{EM}|q_{l-}\rangle|^2}{\sum_l a_l \int \frac{\mathrm{d}^3 q}{2\pi^3}\, \delta(E_{l-}(q) - E_{l1} + \hbar\omega)\, \sum_i |\langle s_{li}|\hat{H}_\mathrm{EM}|q_{l-}\rangle|^2},
\end{aligned}
\quad (4)
$$

where the summation over $l$ is the summation over the 24 Weyl cones of TaAs. $\omega$, $\mathbf{k}_\mathrm{p}$, $\boldsymbol{\varepsilon}$, and $I$ are the frequency, momentum, polarization, and intensity of the pump light entering the material. $|q_{l-}\rangle$ is the spin state of the Weyl band; $v_{l-}(q)$ is the velocity and

$E_{l-}$ is the energy of the Weyl band; $E_{l1}$ is the energy of the massive band, which is assumed to be independent of momentum; $|s_\mathrm{i}\rangle$ is the spin state of the excited massive band, and $\tau_l$ is the current relaxation time. The relaxation time appears in Eq. (4) because Fermi's golden rule yields the number of transitions per unit time and we have to integrate it over the lifetime of the current. The prefactor of $\frac{I}{\hbar\omega}$ is the flux of photons entering the material. The factor $a_l$ is the spatial overlap between the wave functions of the Weyl band and the excited band. We assume that the difference between $E_{l-}(q)$ and the Fermi energy is much greater than the temperature. We neglect the momentum transfer from light to the quasiparticle owing to the small incident angle $\Theta$.

Further details of the calculations can be found in Supplementary Note 4. After performing the integrals, Eq. (4) will be of the form

$$J^\mathrm{i}(\omega, \mathbf{k}_\mathrm{p}, \boldsymbol{\varepsilon}) = \frac{-eI}{\hbar\omega} \frac{\sum_l \tau_l a_l \chi_l N_{(l)j}^\mathrm{i} L^\mathrm{j}}{\sum_l a_l D_{(l)}^\mathrm{ij} \varepsilon_\mathrm{i} \varepsilon_\mathrm{j}^*}, \quad (5)$$

where $N_{(l)j}^\mathrm{i}$ and $D_{(l)}^\mathrm{ij}$ are tensors that depend on the dispersion relations of the cones, and are independent of the frequency of light as long as the Weyl bands are linear; $\chi_l = \pm 1$ is the chirality of each Weyl cone (the $+$ and $-$ signs correspond to right- and left-handed cones, respectively); $\mathbf{L} = \mathrm{i}\boldsymbol{\varepsilon}\times\boldsymbol{\varepsilon}^*$ is the angular momentum per photon. For circularly polarized light, $\mathbf{L} = \pm\hbar\hat{\mathbf{k}}_\mathrm{p}$. Explicit expressions for the tensors $N_{(l)j}^\mathrm{i}$ and $D_{(l)}^\mathrm{ij}$ are given in the Supplementary Note 4.

TaAs has tetragonal symmetry, i.e., fourfold rotational symmetry about an axis and reflection symmetry about four planes containing that axis. It also has time reversal symmetry. This means the 24 Weyl points exist as a set of 8 ($W_1$) and a set of 16 ($W_2$), with the cones in each set related by the crystal symmetries. Chirality is invariant under rotations and time reversal, and flips sign under reflections. This means each set has an equal number of left and right-handed cones. If we take the sum over a set of cones, the symmetric components of $N_{(l)j}^\mathrm{i}$ cancel owing to the tetragonal symmetry of the crystal and the only possible non-canceling contribution is from $N_{(l)y}^\mathrm{x} - N_{(l)x}^\mathrm{y}$. Therefore, the photocurrent in Eq. (4) is $\mathbf{J} \propto \pm\hat{\mathbf{k}}_\mathrm{p}\times\hat{\mathbf{c}}$, where $+$ and $-$ signs refer to the right- and left-handed polarizations of light. $N_{(l)y}^\mathrm{x} - N_{(l)x}^\mathrm{y}$ is non-zero only if the tilt Hamiltonian $\hat{H}_\mathrm{t}$ is non-zero, the untilted part of the Hamiltonian $\hat{H}_\mathrm{W}$ is anisotropic, and the tilt is not

aligned with principal axes of the untilted part (the vector $v_t^i$ is not along any of the principal axes of the tensor $v_a^i v_a^j$).

This photocurrent is perpendicular to both the [001] crystallographic axis and the momentum of light $\mathbf{k}_p$, and it reverses sign for different circular polarization or opposite direction of $\hat{\mathbf{c}}$ axis. The chiral nature of Weyl cones cannot contribute to the LPGE and hence produce a current along the [001] axis. We consider the light incident approximately normal to the (112) face of the crystal. In this case, the chiral photocurrent is along the [$\bar{1}$10] direction.

For excitation light with a wavelength of 800 nm, the numerical evaluation of Eq. (4) yields a value of $J_1 = +940$ nA m$^{-1}$ for the contribution of the eight Weyl cones $W_1$ to the sheet current density. For the classification of the Weyl cones with different chiralities in TaAs, we follow the supplementary material of ref. [25] and use an optical penetration depth of ~25 nm. For the 16 Weyl cones $W_2$, the sheet current density $J_2 = -1340$ nA m$^{-1}$. Owing to the lack of detailed information about the probabilities of excitation for the two sets of cones $W_1$ and $W_2$, we are only able to reliably obtain the range of the sheet current density from $-1340$ to $940$ nA m$^{-1}$. If we assume that 8 Weyl cones $W_1$ and 16 Weyl cones $W_2$ are excited with equal probabilities, we obtain $-580$ nA m$^{-1}$ (for right circular polarization) or $+580$ nA m$^{-1}$ (for left circular polarization). Experimentally, the peak value of the helicity-dependent $J_x(t)$ reaches almost 350 A m$^{-1}$. Considering the photocurrent flows over a time $<\tau_l> \simeq 400$ fs at a repetition rate of $f_{rep} = 1$ KHz, we evaluated an equivalent DC sheet current density via $\overline{J_x} \simeq$ max $(J_x)f_{rep} <\tau_l>$, which gives $\overline{J_x} \sim 140$ nA m$^{-1}$. This value is consistent with the our theoretical result, although there exists some discrepancy between their numbers, which is understandable because our model is based on some assumptions mentioned above.

We found that when the photon energy ($\hbar\omega$) is tuned into the spectral regime that involves excitation of the linear Weyl bands, the sheet photocurrent density will approach to the maximum. Specifically, in this regime $J(\omega)$ follows the relationship: $J(\omega) \propto 1/\omega$. Our calculations show that this $J(\omega) \propto 1/\omega$ regime overlaps with the near-infrared spectrum (close to 1.5 eV). As shown in Fig. 7b, the frequency-dependent experimental data, $\overline{J_x}(\omega)$, for CPGE agree well with the results obtained by our theory. Therefore, our studies elucidate the anisotropic chiral Weyl bands are crucial for generating the giant CPGE-induced photocurrent even in the near-infrared regime.

We note that $J_x(\omega)$ here maximizes at $\hbar\omega \sim 1.5$ eV, which is approximately twice the fundamental resonance frequency at which the second harmonic generation peaks[33]. The frequency-dependent giant SHG in ref. [33] is tentatively explained by the shift current response (LPGE) owing to the skewness of the polarization distribution function in the ground state. Based on the SHG studies, it is still unclear whether their observations are related to the chiral Wely fermions. Moreover, as our theoretical model only deals with the CPGE and cannot apply to the shift current, we are also unable to access the relevance of the SHG to our results.

The observed photocurrent response is very strong and can be verified by the Glass coefficient (G), as done in ref. [26]. For instance, at 800 nm we can obtain G to be ~10 cm V$^{-1}$ and ~ $4 \times 10^{-9}$ cm V$^{-1}$ corresponding to max($J_x$) and $\overline{J_x}$, respectively. The former is enormously huge, and larger than all the known values[26] by a factor of $\sim 10^9 - 10^{10}$. Such giant value directly reflects an ultrafast current pulse with ps timescale. Indeed, the latter, corresponding to the DC photocurrent, is also above the average G value for a large collection of materials in the near-infrared region[26]. We note that our calculated Glass coefficient corresponding to the equivalent DC photocurrent is still

significantly smaller than that associated with the shift current in ref. [26]. Nevertheless, our obtained G value is reasonable given the much higher photon energies employed here.

One would argue that helicity-dependent photocurrents may arise from other mechanisms, such as the circular photon drag effect (CPDE)[43] or the spin-galvanic effect (SGE)[36]. The CPDE current requires additional transfer of the light momentum along the charge current direction. This effect will be irrelevant for the case here with a small incidence angle. For the SGE current, its decay is determined by the spin relaxation time, $\tau_s$[36]. Based on our time-resolved Kerr rotation measurement, the observed $J_x(t)$ cannot be explained by SGE, as $\tau_s \simeq 60$ fs is too small (see Supplementary Fig. 13).

LPGE, on the other hand, depends on the crystal symmetry, manifested by dependence on the linear polarization state of the light. Its induced charge current is generally called the shift current[26,36,40], which occurs when the electron density distribution of the excited state is spatially shifted with respect to the initial states during the optical transitions. LPGE is absent for systems with inversion symmetry under a quadratic nonlinear optical process[36] and is thus also referred to as the bulk photovoltaic effect[26]. Our experimental data are well explained by the phenomenological Eq. 10 derived from this effect. Here, because the Weyl cones dominated by the Ta 5$d$ orbitals and the high-lying bands include both Ta 5$d$ and As 4$p$ orbitals[17], the associated ultrafast electron transfer should occur along both Ta-(As)-Ta and Ta-As bonds (see Fig. 8a). Clearly, the crystal symmetry is embedded in the anisotropy of these bonds. By contrast, for low photon energies, e.g., ~120 meV, the electron transfer directly between Ta and Ta along the Ta-(As)-Ta bond dominates the shift currents. The colossal shift photocurrents found in ref. [26] should belong to this case. Similarly, we can get the G to be ~3 cm V$^{-1}$ and ~$10^{-9}$ cm V$^{-1}$ corresponding to the maximum ultrafast and equivalent DC LPGE-induced currents at 800 nm. These values are also colossal.

Moreover, we measured the photon-energy-dependent LGPE response (see Fig. 8b), which, however, behaves quite differently with that of the CPGE-induced current. In specific, instead of showing resonance behavior around 1.5 eV, $\overline{J_x}(\omega)$ arising from the LPGE seems to show a plateau-like behavior in the near-infrared regime. This result also differs from the frequency-dependent SHG signals[33]. Furthermore, it can hardly be explained by the THz emission observed in the ferroelectric semiconductors with well-defined energy gaps[30,31], because no strong absorption feature in the mid- and near-infrared regimes was reported in the semimetal TaAs (see $\hbar\omega$-dependent absorption coefficient in the Supplementary Fig. 1). Theoretically, it is quite challenging to calculate accurately the frequency-dependent LPGE at high-photon energies, and we leave it for future studies. We note that the difference between LPGE and CPGE might arise from two factors: (1) Independence on the relaxation time $\tau$ for LPGE is usually more reliable. (2) LPGE depends on the berry connection, whereas CPGE relies on the berry curvature. As a result, it is conceivable that the CPGE-induced photocurrent can be larger in our measurements. Nevertheless, our experiments demonstrate that the LPGE-induced ultrafast photocurrent can also be significant, and provide an additional control degree of freedom for the broadband THz pulses using linearly polarized light.

$J_{yz}$ is largely independent of any pump polarization. Therefore, discussing its mechanism can basically exclude the CPGE and LPGE, as they only show a little contribution to the signal. Based on Eq. (1), the dominance of $D_{yz}$ in the $S_{yz}$ (or $E_{yz}$) signal indicates $J_{yz}$ has a thermal origin. Its relatively slow response in the time domain further suggests that scattering processes

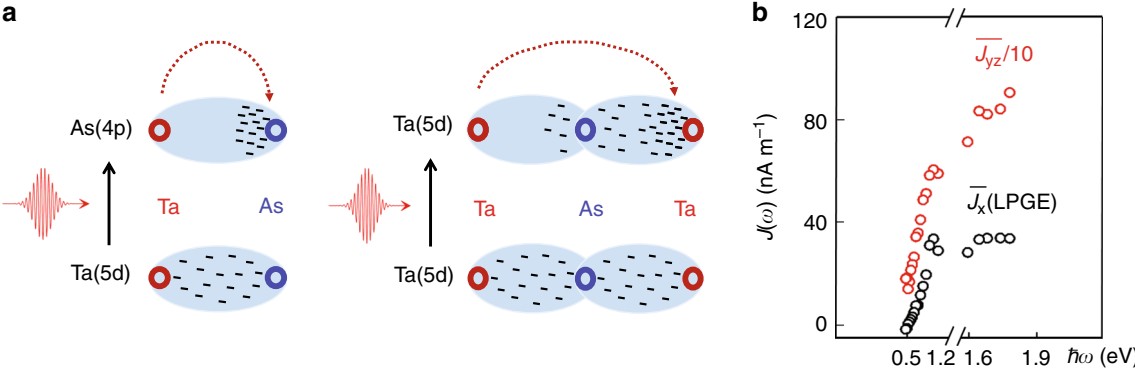

**Fig. 8 Schematic of LPGE and its dependence on the incident photon energy. a** Owing to the initial states being dominated by Ta 5d orbitals and the final states contributed by both Ta 5d and As 4p orbitals[17], the shift current is attributed to the ultrafast transfer of electron density along both the Ta-As and Ta-(As)-Ta bonds. **b** Experimental DC sheet current densities, $\overline{J_x}(\omega)$ and $\overline{J_{yz}}(\omega)$, as a function of the photon energy originating from the LPGE and photo-thermal effects, respectively.

following the initial optical transitions are involved during the generation of $J_{yz}$. Possible candidates are the photo-Dember effect[44] and carrier drift, both of which depend strongly on the phonon or impurity scatterings. In fact, the photon-energy dependent data in Fig. 8b shows that $J_{yz}(\omega)$ continuously increases with $\hbar\omega$. This behavior is very similar to that of the light absorption coefficient $\alpha(\omega)$ (see Supplementary Fig. 1), and further confirms its thermal origin. A rough estimation of $J_y(t)$ and $J_z(t)$ can also be obtained using the data by varying the incident angle $\Theta$ (see Supplementary Fig. 15). Their magnitudes are close to that of the shift current.

Therefore, the distinct polarization-dependent $J_x(t)$ and $J_{yz}(t)$ arise from different physical mechanisms: nonthermal and thermal, respectively. A phase difference between these two components is expected. As a result, the chiral ultrafast photocurrent $\mathbf{J}(t)$ emerges, as evidenced by our experiment. Such phase difference directly determines the observed $\Delta\varphi$ between $E_x(t)$ and $E_{yz}(t)$. An estimation of $\Delta\varphi$ can be roughly made by $\Delta\varphi \simeq \Omega \langle\tau_{ep}\rangle$, where $\Omega$ and $\tau_{ep}$ are the angular THz frequency and the electron–phonon scattering time, respectively. Using the frequency (~1.8 THz) at the peak magnitude of the Fourier transform and the average $\langle\tau_{ep}\rangle$ value of ~400 fs, we obtain $\Delta\varphi \simeq 1.4\pi$, which is close to the values measured values, i.e., $4\pi/3$ for (112) and $3\pi/2$ for (011). Potential anisotropy of the detail electron–phonon scatterings corresponding to different faces may cause such various phases.

The demonstrated generation of chiral ultrafast photocurrents in WSM TaAs offers unique opportunities for novel THz emission with polarization control. The theory underlying the CPGE-induced current is insightful, predicting the photon-energy dependence and demonstrating the essential role of chiral Weyl fermions, in spite of the transition involving a massive band, where there is no clear notion of chirality. In terms of the applications, the simplicity of polarization control of the ultra-broadband THz wave is extremely powerful and effective. Other advantages of the WSM THz emitter include the low cost in sample preparation and the high THz emission efficiency. We further believe that our observation will benefit the study of other novel phenomena led by the Weyl physics, such as the quantized CPGE[24], and the Weyl-orbit effect[45].

## Methods
**Sample details**. Large-size high-quality TaAs crystals with regular shapes and shiny facets were grown by the chemical vapor transport (CVT) method. High-purity elemental tantalum, arsenic, and iodine with a molar ratio of 1:1:0.05 were filled into a silica ampule under argon. The ampule was evacuated to a pressure below 1 Pa, sealed quickly by flame to avoid the loss of iodine and arsenic, and then heated gradually from room temperature to 1000 °C to get TaAs polycrystalline.

Afterwards, the ampule was put to a temperature gradient from 1020 °C to 980 °C where the CVT proceeded for 2 weeks and single crystals were obtained. More details can be seen elsewhere[46]. The crystal structure and orientations were determined by x-ray diffraction method and the average stoichiometry was confirmed by energy-dispersive x-ray spectroscopy. TaAs crystallizes in a body-centered tetragonal unit cell where Ta and As atoms are six coordinated to each other. The corresponding lattice constants are $a = b = 3.43$ Å and $c = 11.64$ Å, and the space group is I41 md (No. 109).

**Experimental setup**. In our experiments, the sample was excited by the laser pulses from a Ti:sapphire amplifier (repetition rate 1 KHz, duration ~80 fs, and center wavelength 800 nm) under ~3° or 45° angle of incidence. Other wavelengths of the excitation light come from an optical parametric amplifier, which produces similar pulse duration (~80 fs). The beam diameter is ~1.5 mm (full-width at half intensity maximum). A typical pump power of 25 mW was used. The THz electric field was detected by EO sampling, with probe pulses from the same laser co-propagating with the THz field through an EO crystal, which is the ZnTe(110) with a thickness of 0.4 mm. Measurements were performed at room temperature in a dry-air environment with relative humidity <5%. All data were collected in the linear regime, i.e., amplitude of THz field increases linearly with the pump power (see Supplementary Fig. 5).

A quarter-wave ($\lambda/4$) or half-wave ($\lambda/2$) plate mounted in a computer-controlled rotation stage is employed to tune the polarization state of the optical pulses just before they reach to the sample. A THz wire-grid polarizer (field extinction ratio of $10^{-2}$) allows us to measure the s and p components $E_x$ and $E_{yz}$ of the THz electric field separately, thereby disentangling current components $J_x$ and $J_{yz}$. The latter is a linear combination of $J_y$ and $J_z$[47].

**Extraction of the THz electric fields**. To extract the emitted THz electric field $\mathbf{E}(t)$ directly above the sample surface from the measured EO THz signal $\mathbf{S}(t)$, there is a linear relationship between these two quantities[34]. For instance, in the frequency domain, the THz field component $E_x$ and the corresponding signal $S_x$ are connected by the total transfer function $h(\Omega)$ through the simple multiplication[34]

$$S_x(\Omega) = h(\Omega)E_x(\Omega), \quad (6)$$

where $h(\Omega) = h_{det}(\Omega)h_{prop}(\Omega)$ includes the detector response $h_{det}(\Omega)$ and the transfer function $h_{prop}(\Omega)$ of the THz wave from the sample to the detection crystal. The same relationship is valid for $E_{yz}$ and $S_{yz}$. Details of the transfer functions are shown in the Supplementary Fig. 14.

**Extraction of the ultrafast photocurrents**. In order to obtain the source current $\mathbf{J}(t)$ from the THz electric field $\mathbf{E}(t)$ measured directly above the sample surface, we make use of the following generalized Ohm's law[39,47]:

$$E_x(\Omega) = -\frac{Z_0}{\cos\Theta + \sqrt{n^2 - \sin^2\Theta}}J_x(\Omega) \quad (7)$$

$$E_{yz}(\Omega) = -\frac{Z_0 \sin\Theta}{n^2 \cos\Theta + \sqrt{n^2 - \sin^2\Theta}}J_{yz}(\Omega). \quad (8)$$

Here, $\Omega$ is the THz frequency, $Z_0(\simeq 377$ Ohm) is the vacuum impedance, $n$ is the refractive index of TaAs at THz frequency (see Supplementary Fig. 2), $\Theta$ is the angle of incidence, and

$$J_{yz} = J_z(\Omega) - \frac{\sqrt{n^2 - \sin^2\Theta}}{\sin\Theta}J_y(\Omega) \quad (9)$$

is a weighted sum of the currents flowing along the $\hat{\mathbf{y}}$ and $\hat{\mathbf{z}}$ directions. The inverse Fourier transformation of the resulting current spectra yields the currents in the time domain.

**Photocurrents owing to the LPGE**. Phenomenologically, the non-local transient photocurrent $\mathbf{j}(\mathbf{r}, \Omega)$ at THz frequency $\Omega$ owing to the LPGE can be described by a quadratic nonlinear optical process[34,36,39,48]

$$j_\lambda(\mathbf{r}, \Omega) = 2 \sum_{\mu\nu} \int_{\omega>0} d\omega \, \xi_{\lambda\mu\nu}(\mathbf{r}; \omega+\Omega, \omega) F_\mu f_\nu^*, \qquad (10)$$

where $\lambda$, $\mu$, and $\nu$ stand for the Cartesian coordinates $\hat{\mathbf{x}}$, $\hat{\mathbf{y}}$, and $\hat{\mathbf{z}}$. $\xi_{\lambda\mu\nu}$ is the third-rank pseudo-tensor. $\mathbf{F}$ and $\mathbf{f}$ are the complex-valued pump-field Fourier amplitudes at frequencies $\omega + \Omega$ and $\omega$ originating from the fs optical pump pulse. Owing to $\omega \gg \Omega$, $|\mathbf{F}(\omega+\Omega)| \simeq |\mathbf{f}(\omega)|$.

For TaAs with inversion symmetry broken, there are three independent nonvanishing elements of $\xi_{\lambda\mu\nu}$[32]: $\xi_{zzz}$, $\xi_{zxx} = \xi_{zyy}$ and $\xi_{xzx} = \xi_{yzy} = \xi_{xxz} = \xi_{yyz}$. They are defined in the coordinates for (001) face, where $\hat{\mathbf{x}}$, $\hat{\mathbf{y}}$, and $\hat{\mathbf{z}}$ are parallel to the unit cell axises $\hat{\mathbf{a}}$, $\hat{\mathbf{b}}$, and $\hat{\mathbf{c}}$, respectively. For the coordinates of other faces, i.e., (011) and (112), they will follow the transformation of the rotation matrix. Therefore, $j_\lambda(\mathbf{r}, \Omega)$ owing to the LPGE will be different at different faces (see Supplementary Note 1).

## Data availability
The data that support the findings of this study are available from the corresponding author upon reasonable request.

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

## Acknowledgements
This research is supported by the National Natural Science Foundation of China (Grant Nos. 11974070, 11734006, and 11674369), the Frontier Science Project of Dongguan (No. 2019622101004), the National Key Research and Development Program of China (Nos. 2016YFA0300600 and 2018YFA0305700), the Science Challenge Project of China (TZ2016004), the National Postdoctoral Program for Innovative Talents (Projects No. BX201700012, funded by China Postdoctoral Science Foundation), and the CAS interdisciplinary innovation team. M.K.L. and X.C. acknowledge support from National Science Foundation under Grant No. DMR-1904576. This work is also

supported in part by the US Department of Energy under Awards DE-SC-0017662 (S.K. and D.E.K.) and DE-FG-88ER40388 (E.J.P. and D.E.K.).

## Author contributions

J.Q. conceived the project, and designed the experiments. J.Q. and M.K.L. supervised this work. Y.G. built the THz emission setup and performed the measurements with help of Y.L., W.L.Z., Y.S. and X.C. Z.L. prepared the TaAs sample. Y.G., Y.Q., M.K.L. and J.Q. analyzed the experimental data. S.K., E.J.P. and D.E.K. provided theoretical description of the data and performed theoretical computation of the helicity-dependent photocurrents. M.K.L. and H.W. provided theoretical assistance. Y.G., M.K.L. and J.Q. wrote the manuscript with contributions from all the co-authors. Y.G., S.K., E.J.P. and Z.L. contributed equally to the completeness of this work.

## Competing interests

The authors declare that they have no competing interests.
