## [Peer Review File · Nature Communications]

Reviewers' Comments:

Reviewer #1:

Remarks to the Author:

The authors report a comprehensive study of the THz emission from a Weyl semimetal. This is very timely given the interest in the nonlinear response of these materials as well as the development of new tools to probe their chirality. Particularly exciting is the possibility of creating a new source of THz circularly polarized light. Furthermore, there is an extensive data set that potentially provides important insights into this field, and answers important questions. As such this paper could potentially be published in Nature Communications.

First the very positive aspects of the paper:

The study of the wavelength dependence of the THz/photocurrent generation is quite important and provides new insights, though not entirely highlighted by the paper (see below). Additionally, the authors' careful study of the symmetry response clearly isolates the thermal from non-thermal responses, very convincingly shows which terms are really due to intrinsic effects.

However, before publication a number of issues need to be resolved, some major and some minor.

1 - Throughout the manuscript, the authors refer to "control" of THz chirality on an ultrafast time scale. In fairness, it is not really clear this is achieved. They certainly can generate circularly polarized THz, but they do not really control it on a fast time scale. Indeed, as far as I can tell it's not really clear they get anything better than putting a Fresnel Rhomb in front of a standard THz source. I would suggest either rewording in terms of an intrinsic source of helical light or really showing they can switch the polarization on ultrafast time scales (frankly the second would be fantastic but perhaps better saved for the next paper).

2 - Related to the above, it is not really clear how "ultrafast" this is. In some parts of the paper, this is discussed in reference to Figure 6, but I don't really understand how the rise and decay time scales are extracted from this. Perhaps this is not so crucial compared to the broadband response?

3 - The authors have left two important references, namely the recent work of the LANL group published in PRL on TaAs and the work of Ogawa on shift currents in Ferroelectrics and TI's. It would be important to explain what is new here, such as the energy dependence (though Ogawa did this on TI).

4- Related to comparing to other works, there is some discussion of the 2 ω work of the Berkeley/temple group. What is quite nice here is the demonstration of similar resonance, however it seems the resonance observed here is at the 2 ω of where the SHG peaks. This combined with the clear demonstration that this is connected to the node is an important advance, that was unclear in the SHG. It suggests the SHG peak is not really due to transitions/resonance at ω but rather at 2 ω .

5- The authors often emphasize the size of the current generated, however one should really compare the responsivity of better yet the intrinsic second order terms. For example achieving 10 times the current but with 100 times the fluence would not be so impressive. For example how does the Glass coefficient measured here compare with the work of Ref 29?

6- A minor point, but related to the above, Ref 29 has now been published in Nat. Materials, so it should be updated.

7 - The theory discussion is a bit hard to follow. It seems to reformulate things quite differently than previous CPGE and BPVE. For example, what are the N and D tensors? Also the claim that this mechanism is different than previous works is not correct. For example the process shown in Fig 7a, without tilt would give zero as the opposite cone of opposite chirality would give the opposite

CPGE.

8 - related to the above, I don't understand the estimate of the phase difference from the frequency and scattering time.

9 - lastly the claim about linear in ω being extraordinary is far overblown. First the data is far too sparse to support this. Second even if true why would this be a big deal? I believe the point is to explain the relevance of linear versus nonlinear terms of the Weyl band dispersion. That would be interesting, but not really extraordinary, especially given how little is known about the final state and frankly, I don't really see how the nonlinear response is really all that different in the two regimes. Perhaps it is worth removing this claim and focusing on the explanation of the peak in the response and its connection to former SHG...

Reviewer #2:

Remarks to the Author:

The manuscript by Gao et al. presented a systematic study of the terahertz wave emission from the Weyl semimetal TaAs.

The general introduction is well-written. The authors conveyed well the excitement and significance of the ultrafast responses of Weyl semimetals. The experiments are well carried out and the main data observations are well described. Based on general excitement on nonlinear responses of Weyl semimetals and topological materials, I recommend the publication of this manuscript if the authors can properly address the following questions.

1. Ref. 30 also reported ultrafast THz emission in the Weyl semimetal TaAs. Some of the key measurements are similar. Therefore, the authors need to clearly and properly cite that paper (currently it is incorrectly cited as "time-averaged" measurements). More importantly, the authors should clearly describe how the present work is different or goes beyond Ref. 30.
2. Theoretically, the shift current and the CPGE should have different dependence on the relaxation time τ . The CPGE is proportional to τ^1 whereas the shift current is proportional to τ^0 (i.e. independent of τ). The authors should discuss whether they observe anything related to that.

Reviewer #3:

Remarks to the Author:

Gao et al. present the study of THz emission induced by ultrafast light in the prototypical Weyl semimetal TaAs. Despite the intensive interest lately in this topic, I don't recommend publication of the paper (at least in its current form) in Nature Communications for the following reasons:

1. Over the past couple of years, there have been quite a few studies in the light-induced current generation in Weyl semimetals and different mechanisms have been discussed. These studies include both steady-state measurements with continuous-wave lasers and time-resolved measurements with femtosecond pulsed lasers. The authors have cited most of them in the reference list. Compared to the previous publications on this topic (Ref. [27-30] cited by the authors), I didn't see Gao et al. presents enough novelty to advance our understanding of this topic. Particularly, compared to Ref. [30] that presented very similar studies, the authors failed to summarize what has already been observed and analyzed and what is new here.

For instance, in the beginning, the authors mentioned "However, up to now there is very little information about the ultrafast photocurrent in WSMs, except that its time-averaged direction can be derived by considering the crystal symmetry [27, 30]. Particularly, properties of the ultrafast

photocurrent induced THz emission and its associated Weyl electron dynamics are still unclear.” Ref.[30] is not time-averaged measurement and it reveals very similar dynamics as the current paper.

From my point of view, the main advance of this paper (compared to Ref. [30]) is its quantitative computation of the injection current based on the real band structures of TaAs. This quantitative computation is very complicated due to the high photon energy used. Does the schematic in Fig. 7a reflect the real situation? Why is there no excitation from lower bands (occupied) to the Weyl bands above the Fermi level (unoccupied)? Moreover, the authors argued that the mechanism is different from the previous studies with lower photon energy that connects only Weyl bands. I don't understand such an argument. Both are due to the combination of optical selection and the asymmetric bands. The band velocity difference is always there if only considering a one-photon excitation event. Even for a perfect Dirac cone as in graphene, if we only consider one excitation event by one photon, the electron has opposite velocities before and after being excited. The total current may be canceled out by considering the space-time symmetry, which is reflected in the symmetry and angular momentum index of the bands.

2. Another possible novel aspect of this paper is the excitation photon energy dependence, as mentioned in their abstract “The photocurrent generation is maximized at near-IR frequency range close to 1.5 eV”. However, this was not given sufficient discussions in the paper. Since the authors talk about all CPGE (injection current), LPGE (shift current) and thermal current generation, which photocurrent is maximized at 1.5 eV? Are they all enhanced at the same excitation photon energy? PRB 98, 165113 (2018) reported a resonant enhancement of second harmonic response at ~ 1.5 eV (with ~ 0.7 eV as the principle excitation). Do the authors see similar photon energy dependence for the shift current?

In summary, due to the above concerns and questions, I am reluctant to recommend its publication in Nature Communications.

Minor aspects:

1. A few references need to be updated, such as [29] and [30].
2. Some parts of the presentation confused people about the role of the crystal symmetry in the generation of LPGE and CPGE. For instance, “LPGE, on the other hand, depends on the crystal symmetry or the linear polarization of light”. Both LPGE and CPGE, similar to SHG, as long as they are second-order, should be governed by the crystal symmetry.

Response to Reviewer #1:

We thank the referee for careful reviewing our manuscript and being supportive of the publication of our manuscript in Nature Communications. In the revised manuscript, we include responses to his/her concerns. Below, we list the referees' comments in blue italics and our responses in normal font. At the end of this document, we have provided a summary of changes.

The authors report a comprehensive study of the THz emission from a Weyl semimetal. This is very timely given the interest in the nonlinear response of these materials as well as the development of new tools to probe their chirality. Particularly exciting is the possibility of creating a new source of THz circularly polarized light. Furthermore, there is an extensive data set that potentially provides important insights into this field, and answers important questions. As such this paper could potentially be published in Nature Communications.

First the very positive aspects of the paper: The study of the wavelength dependence of the THz/photocurrent generation is quite important and provides new insights, though not entirely highlighted by the paper (see below). Additionally, the authors' careful study of the symmetry response clearly isolates the thermal from non-thermal responses, very convincingly shows which terms are really due to intrinsic effects. However, before publication, a number of issues need to be resolved, some major and some minor.

We thank the referee for praising the importance of this work, especially the wavelength-dependent data. We have substantially modified our manuscript according to his/her comments and suggestions. We have especially highlighted the wavelength-dependent part of the work.

1 - Throughout the manuscript, the authors refer to "control" of THz chirality on an ultrafast time scale. In fairness, it is not really clear this is achieved. They certainly can generate circularly polarized THz, but they do not really control it on a fast time scale. Indeed, as far as I can tell it's not really clear they get anything better than putting a Fresnel Rhomb in front of a standard THz source. I would suggest either rewording in terms of an intrinsic source of helical light or really showing they can switch the polarization on ultrafast time scales (frankly the second would be fantastic but perhaps better saved for the next paper).

We thank the referee for pointing out that we should clarify what “control” means in our work. As the referee suggested, we removed the phrase “ultrafast control” in the revised manuscript.

We used the term “control” to state that the polarity of the emitted THz wave can be manipulated via the control of the photocurrents upon femtosecond laser excitation

with well-defined polarization. The Fresnel Rhomb mentioned by the referee is a common way to manipulate polarization of the light in an arbitrary manner. Although it works well at the visible to mid-infrared range, it yields little or no practical use for the ultra-broadband THz wave manipulation, e.g. λ between ~ 100 and $3000 \mu\text{m}$. In fact, there are no such commercial products available. The existing THz waveplate is either for single color or has a narrow bandwidth (<1 THz). That is why control of the ultra-broadband THz polarization is one of the most challenging topics in the THz community [Phys. Rev. Lett. 92, 208301 (2004), Nat. Photonics 7, 724 (2013), Nat. Photonics 12, 554 (2018)].

Previous works (mentioned above) realizing the polarization control of the ultra-broadband THz wave are far more complicated than our method. In this work, we show that, using one Weyl semimetal, THz wave can be generated with different elliptical polarization, which is not possible with conventional nonlinear crystals and photocurrent emitters. Therefore, the Weyl semimetal can be an intrinsic helical broadband THz wave source. We believe this work will certainly attract the attention of many researchers focusing on the applications of the topological chiral materials.

2 - Related to the above, it is not really clear how "ultrafast" this is. In some parts of the paper, this is discussed in reference to Figure 6, but I don't really understand how the rise and decay time scales are extracted from this. Perhaps this is not so crucial compared to the broadband response?

First, as the referee mentioned, due to the broadband response of THz emission, we estimate the timescale of "ultrafast" using the measured THz bandwidth, i.e. $12 \text{ THz} \sim 80 \text{ fs}$, which basically determines how fast the rise time is in the experiment. The decay time scale is evaluated by the time needed for the peak current density dropping to a value by a factor of $1/e$.

Second, the word "ultrafast" prior to "control" denotes that switching of the key components of the photo-induced currents, i.e. the component due to CPGE or LPGE, does not rely on the laser-induced heating effect (non-thermal origin). With a timescale of $\sim 80 \text{ fs}$, we can realize the photocurrent switching using the fs optical pulses with various polarizations. The case here is the same as a recent work reporting the THz emission from magnetic heterostructure systems [T. J. Huisman et al., *Femtosecond control of electric currents in metallic ferromagnetic heterostructures*, Nat. Nanotech. 11, 455(2016)]. Similar usage of "ultrafast control" is also widely seen in the research of spin or magnetization control by femtosecond light [A. Kirilyuk et al., *Ultrafast optical manipulation of magnetic order*, Rev. Mod. Phys. 82,2731 (2010)].

3 - The authors have left two important references, namely the recent work of the LANL group published in PRL (Ref. 30) on TaAs and the work of Ogawa on shift currents in Ferroelectrics and TI's. It would be important to explain what is new here,

such as the energy dependence (though Ogawa did this on TI).

We have cited the LANL work published in PRL [30] (Ref.[29] in the revised manuscript). We used the arXiv number because by the time of the submission that paper was not in print yet. In the revised manuscript, we have updated this reference, added Ogawa's very recent papers on ferroelectric semiconductors [Proc. Natl. Acad. Sci. 116, 1929 (2019); Appl. Phys. Lett. 114, 151101 (2019)], and have clarified the new findings as well. The novelty of our work is in three-fold:

(1) *Analysis of the crystal symmetry via group theory and the detail band topology/structures* are both crucial for determining the photocurrents having nonthermal origins (CPGE and LPGE). The former tells the existence of these photocurrents, and the latter determines their associated amplitudes. Previous work in Ref. [30] only using ~ 1.5 eV photon addressed the former issue in TaAs. Because the latter issue, directly connected to the underlying physics, is much more complicated, it is still unknown. In order to explore this problem, quantitatively obtaining the photocurrents and measuring the photon-energy dependence are mandatory.

In semimetal TaAs, we, for the first time, performed experiments over a broad spectral range (covering mid- and near-infrared light), and provided exact amplitudes of the ultrafast photocurrents. Both of these aspects were not discussed in LANL's work (which was not published at the time of our submission). We experimentally demonstrated that a resonance behavior resides near ~ 1.5 eV which yields the most efficient photocurrent generation within the investigated photon energies.

These new results together with our theoretical calculations enable us to reveal how the peculiar band structures near the Weyl nodes affect the photocurrent generation.

(2) We have theoretically calculated the generated CPGE photocurrent as a function of incident light wavelength upon excitation of high photon energies in TaAs for the first time. Our work reveals that the low energy excitations – the chiral fermions in the Weyl bands near E_f - surprisingly dominate the giant photocurrent response even when the photon energy is very high (in the near-infrared regime). This is mainly attributed to the excitation between the anisotropic chiral Weyl band and the massive bulk band accompanied by a large and rapid change in the effective velocity of the charged quasiparticle, and hence by a giant coherent THz radiation. Accuracy of our calculations is confirmed by the experiments: a) The calculated photocurrent amplitude consists well with the experimental data. b) We predict that as the photon energy is tuned away from ~ 1.5 eV, the photocurrent will decay fast, precisely as evidenced by our experiments.

(3) We demonstrate that it is extremely powerful and convenient to control the elliptical polarization of **ultra-broadband** THz wave for the Weyl semimetal TaAs. We have directly obtained the correct bandwidth and phase of the near-field THz

wave (or ultrafast photocurrent), which are vital for extracting the ultrafast photocurrents and future THz applications. All these features were not disclosed in previous work [30]. We discover that this type of THz emitter has many superior advantages, such as large electric field comparable to the commercial THz nonlinear crystals, simple and arbitrary elliptical polarization control with ultra-broadband width and high dynamical range. These findings for the first time provide an exceptional example of the Weyl semimetals being used in a practical application, and also open a new route to realize chiral photon sources using quantum materials.

(4) In the works by Ogawa et al., the authors mainly discuss the shift current response in ferroelectric semiconductors. However, we here mainly focus on discussing the dominant CPGE-induced photocurrents in semimetals. Although the photon-energy dependent data have similarities between them, the photocurrents due to LPGE (shift current) and CPGE, microscopically, have very different mechanisms. Because at this stage we do not have a microscopic theory for LPGE in WSMs, we leave the detail discussion of LPGE for future work.

4- Related to comparing to other works, there is some discussion of the 2omega work of the Berkeley/temple group. What is quite nice here is the demonstration of similar resonance, however it seems the resonance observed here is at the 2omega of where the SHG peaks. This combined with the clear demonstration that this is connected to the node is an important advance, that was unclear in the SHG. It suggests the SHG peak is not really due to transitions/resonance at omega but rather at 2omega.

We thank the referee for pointing out the important advance of our work. The photon-energy dependent current $J_x(\omega)$ due to CPGE maximizes at ~ 1.5 eV, which is approximately twice the fundamental resonance energy at which the second harmonic generation peaks [S. Patankar et al., PRB, 98, 165113 (2018)]. The photon-energy dependent giant SHG in Ref. [S. Patankar et al., PRB, 98, 165113 (2018)] is tentatively explained by the nonlinear shift current response due to the skewness of the polarization distribution function in the ground state. Based on those SHG studies, it is still unclear whether their observed behavior is related to the chiral Weyl fermions or not. Although investigations in our work elucidate that the chiral Weyl nodes are responsible for the giant CPGE-induced photocurrent in the near-infrared regime, we are unable to access the relevance to the SHG to our observations since our theoretical model only deals with the CPGE and cannot apply to the shift current. We have included these discussions in the revised manuscript.

5- The authors often emphasize the size of the current generated, however one should really compare the responsivity of better yet the intrinsic second order terms. For example achieving 10 times the current but with 100 times the fluence would not be so impressive. For example how does the Glass coefficient measured here compare with the work of Ref 29?

We agree with the referee that it is better to use the glass coefficient for comparison. We have added the corresponding information into the revised manuscript.

Here we only estimate the Glass Coefficient approximately using the data of J_x component (CPGE) in Fig. 6(a). Using measured penetration depth ~ 25 nm of the 800 nm light (see Supplemental Material), we obtain that the peak pulse current density $\max(J_x)$ and the equivalent DC current density $\bar{J}_x \sim \max(J_x) f_{\text{rep}} \langle \tau_l \rangle$ are $\sim 1.4 \times 10^{10}$ A/m² and ~ 5.6 A/m², respectively. The absorption coefficient α for 800 nm was measured to be $\sim 4 \times 10^7$ m⁻¹. The spot size was estimated by shining the laser through a small calibrated pinhole, yielding a beam diameter of ~ 1.5 mm. The incident power was 25 mW. The Glass coefficient is defined as $G = J/\alpha I$ [29] (Ref. [26] in the revised manuscript), where J is the current density, and I is incident laser intensity.

Using these values, we obtain G to be ~ 10 cm/V and $\sim 4 \times 10^{-9}$ cm/V corresponding to the CPGE-induced peak pulse current density and equivalent DC current density, respectively. The difference between these two values is enormous (nearly by a factor of 10^{10}). This is the main reason we emphasized the giant amplitude of ultrashort current being generated. The value of G estimated using the equivalent DC photocurrent on TaAs at 800 nm is above the average values based on the data in near-infrared region listed for a large collection of materials ([29]). The G values for LPGE is about 4 times smaller than those for CPGE.

6- A minor point, but related to the above, Ref 29 has now been published in Nat. Materials, so it should be updated.

We thank the referee for bringing this up. I have now updated this reference.

7 - The theory discussion is a bit hard to follow. It seems to reformulate things quite differently than previous CPGE and BPVE. For example, what are the N and D tensors? Also the claim that this mechanism is different than previous works is not correct. For example the process shown in Fig 7a, without tilt would give zero as the opposite cone of opposite chirality would give the opposite CPGE.

In revised manuscript, we have presented our theoretical model more clearly. We have added a lot of details including the previously omitted expressions for the tensors N and D .

The referee is right in assuming that there will be no current if there is no tilt. The primary difference from previous calculations, we initially want to claim is that in our work the transition is not from a Weyl band to another Weyl band, where the Pauli blockade plays the essential role in CPGE. The effect seen here is important because it shows that the anisotropic Weyl cone (large band velocity) and its chirality are key factors for the CPGE-induced photocurrent even when the optical transition is

between a linear Weyl band and a massive bulk band (which should not have the concept of chirality) for high photon energies in the near-infrared region.

Here is a short explanation of our model. Unlike the previous works [C.-K. Chan et al., PRB 95, 041104(2017); Q. Ma et al., Nat. Phys. 13, 842(2017)], our theoretical treatment does not rely on the Pauli blockade. Here, the optical selection and asymmetry are important but not the most important. The key factors determining the injection photocurrent density $J_i(\omega)$ at high photon energies ($\hbar\omega$) are the net effect of the tilted anisotropic Weyl cone, its chirality, and the large band velocity difference after excitation. In specific, our model derives $J_i(\omega)$ as [see the main text and Supplemental Material for details]

$$J_i(\omega, \vec{k}_p, \vec{\epsilon}) = \frac{-eI}{\hbar\omega} \frac{\sum_l \tau_l a_l \chi_l N_{(l)j}^i L^j}{\sum_l a_l D_{(l)}^{ij} \epsilon_i \epsilon_j^*} .$$

The above equation has the parameter of chirality χ_l . It also has information about the Weyl cones and band velocities, which are imbedded in the tensors $N_{(l)j}^i$ and $D_{(l)}^{ij}$. If we take the sum over a set of cones with different chiralities, the symmetric components of $N_{(l)j}^i$ cancel due to the tetragonal symmetry of the crystal and the only possible non-canceling contribution is from antisymmetric x-y component $N_{(l)y}^x - N_{(l)x}^y$, which is non-zero only if the tilt Hamiltonian H_t is non-zero, the untilted part of the Hamiltonian H_w is anisotropic, and the tilt is not aligned with principal axes of the untilted part.

8 - related to the above, I dont understand the estimate of the phase difference from the frequency and scattering time.

Estimation of the phase difference between J_x (or E_x) and J_{yz} (or E_{yz}) is actually not from the theoretical calculations because we don't have good models to calculate the shift and photo-thermal currents. The estimation is very rough because we evaluate the average value $\Delta\varphi$ from the dominant THz frequency $\bar{\Omega}$ and the electron-phonon scattering time τ_{ep} by $\Delta\varphi \sim \bar{\Omega}\tau_{ep}$. This equation is based on our finding that J_x and J_{yz} has non-thermal and thermal origins, respectively. The non-thermal part is expected to rise instantaneously. By contrast, rise time of the thermal part is physically determined by the electron-phonon scattering process. The time interval between them roughly gives the phase difference.

9 - lastly the claim about linear in omega being extraordinary is far overblown. First the data is far to sparse to support this. Second even if true why would this be a big deal? I believe the point is to explain the relevance of linear versus nonlinear terms of the Weyl band dispersion. That would be interesting, but not really extraordinary, especially given how little is known about the final state and frankly, I don't really see how the nonlinear response is really all that different in the two regimes. Perhaps it is

worth removing this claim and focusing on the explanation of the peak in the response and its connection to former SHG...

As per the suggestion, we have toned down our claim. In the revised manuscript, we only made a short discussion about these two regimes. As discussed in point 4 above, we have included a brief discussion about the SHG.

Response to Reviewer #2:

We thank the referee for careful reviewing our manuscript and being supportive of the publication of our manuscript in Nature Communications. In the revised manuscript, we include responses to his/her concerns. Below, we list the referees' comments in blue italics and our responses in normal font. At the end of this document, we have provided a summary of changes.

The manuscript by Gao et al. presented a systematic study of the terahertz wave emission from the Weyl semimetal TaAs. The general introduction is well-written. The authors conveyed well the excitement and significance of the ultrafast responses of Weyl semimetals. The experiments are well carried out and the main data observations are well described. Based on general excitations on nonlinear responses of Weyl semimetals and topological materials, I recommend the publication of this manuscript if the authors can properly address the following questions.

We thank the referee for his/her recommendation for publication of our manuscript. In the revised manuscript, we have addressed the questions raised as seen below.

1. Ref. 30 also reported ultrafast THz emission in the Weyl semimetal TaAs. Some of the key measurements are similar. Therefore, the authors need to clearly and properly cite that paper (currently it is incorrectly cited as "time-averaged" measurements). More importantly, the authors should clearly describe how the present work is different or goes beyond Ref. 30.

We clarified what has been done and what has not been done in Ref.[30] and our work. As suggested by the referee, we have modified the corresponding parts in the revised manuscript.

***The main work in Ref. [30] (Ref. [29] in the revised manuscript) are summarized as follows,

(1) *What has been done in Ref. [30]:* Only measuring the far-field THz emission with an inaccurate bandwidth (~0.3-2.2 THz) at 800 nm (~1.55 eV); symmetry analysis via group theory for CPGE; electric-field-induced second-harmonic generation at 800 nm.

(2) What has **NOT** been done in Ref.[30]: Obtaining the near-field THz emission with transform limited bandwidth (0.2 – 12 THz) and complete phase information; providing quantitatively the experimental ultrafast photocurrents; measuring the photon-energy ($\hbar\omega$) dependent THz emission and photocurrent; theoretically revealing the influence of band topology of the Weyl semimetal to the photocurrent.

***We have studied those aspects mentioned above which have not been done in Ref. [30]. The main advances in our work are shown below.

(1) For studying second-order photocurrents (CPGE and LPGE), *analysis of crystal symmetry via group theory and the detail band topology/structures* are crucial. The former tells the existence of these photocurrents, and the latter determines their associated amplitudes. Ref. [30] only addressed the former issue. The latter issue is much more complicated to investigate. In order to explore this problem, quantitatively extracting the photocurrents and measuring the photon-energy dependence must be required.

In semimetal TaAs, we, for the first time, performed experiments over a broad photon energy range 0.5-1.9 eV and provided exact amplitudes of the ultrafast photocurrents. Both of these aspects were not discussed in LANL's work (which was not published at the time of our submission). We experimentally demonstrated that a resonance behavior resides near ~ 1.5 eV which yields the most efficient CPGE-induced photocurrent generation within the investigated spectral range.

These new results together with our theoretical calculations enable us to reveal how the peculiar band structures near the Weyl nodes affect the photocurrent generation.

(2) We have theoretically calculated the generated CPGE photocurrent as a function of incident light wavelength upon excitation of high photon energies in TaAs for the first time. Our work reveals that the low energy excitations – the chiral fermions in the Weyl bands near E_f - surprisingly dominate the giant photocurrent response even when the photon energy is very high (in the near-infrared regime). This is mainly attributed to the excitation between the anisotropic chiral Weyl band and the excited band accompanied by a large and rapid change in the effective velocity of the charged quasiparticles, and hence by a giant coherent THz radiation. Accuracy of our calculations is confirmed by the experiments: a) The calculated photocurrent amplitude consists well with the experimental data. b) We predict that as the photon energy is tuned away from ~ 1.5 eV, the photocurrent will decay fast, precisely as evidenced by our experiments.

(3) We demonstrate that it is extremely powerful and convenient to control the polarization of **ultra-broadband** THz wave for the Weyl semimetal TaAs. We have directly obtained the correct bandwidth and phase of near-field THz wave (or ultrafast

photocurrent), which are vital for extracting the ultrafast photocurrents and future THz applications. All these features were not disclosed in previous work [30]. We discover that this type of THz emitter has many superior advantages, such as large electric field comparable to the commercial THz nonlinear crystals, simple and arbitrary polarization control with ultra-broadband width and high dynamical range. These findings for the first time provide an exceptional example of the Weyl semimetals being used in a practical application, and also open a new route to realize chiral photon sources using quantum materials.

2. Theoretically, the shift current and the CPGE should have different dependence on the relaxation time τ . The CPGE is proportional to τ^{-1} whereas the shift current is proportional to τ^0 (i.e. independent of τ). The authors should discuss whether they observe anything related to that.

The relaxation time τ is an intrinsic property of the crystal, which cannot be varied independently. It appears in our theoretical derivation of the CPGE. However, because we do not have an exact model to calculate the LPGE, we do not know how it exactly connects to the shift current. Both detail band topology/structures and τ will determine the amplitude of the photocurrents. The related calculations will be very complicate. In principle, small τ will lead to small photocurrent. Based on our experiments, the magnitude of the photocurrent via CPGE is about four times larger than that via LPGE. This observation might indicate some difference between their associated relaxation times. But we cannot derive any definite information about them, as mentioned above.

Response to Reviewer #3:

We thank the referee for his/her careful reviewing and positive comments on our work. In the revised manuscript, we include responses to his/her concerns. Below, we list the referees' comments in blue italics and our responses in normal font. At the end of this document, we have provided a summary of changes.

1. Over the past couple of years, there have been quite a few studies in the light-induced current generation in Weyl semimetals and different mechanisms have been discussed. These studies include both steady-state measurements with continuous-wave lasers and time-resolved measurements with femtosecond pulsed lasers. The authors have cited most of them in the reference list. Compared to the previous publications on this topic (Ref. [27-30] cited by the authors), I didn't see Gao et al. presents enough novelty to advance our understanding of this topic. Particularly, compared to Ref. [30] that presented very similar studies, the authors failed to summarize what has already been observed and analyzed and what is new here.

For instance, in the beginning, the authors mentioned “However, up to now there is very little information about the ultrafast photocurrent in WSMs, except that its time-averaged direction can be derived by considering the crystal symmetry [27, 30]. Particularly, properties of the ultrafast photocurrent induced THz emission and its associated Weyl electron dynamics are still unclear.” Ref.[30] is not time-averaged measurement and it reveals very similar dynamics as the current paper.

As pointed out by the referee, we have modified the manuscript. We have clarified what has been done and what has not been done in Ref.[30] and our work.

***The main work in Ref. [30] (Ref. [29] in the revised manuscript) are summarized as follows,

(1) *What has been done in Ref. [30]:* Only measuring the far-field THz emission with an inaccurate bandwidth ($\sim 0.3\text{-}2.2$ THz) at 800 nm; the symmetry analysis via group theory for CPGE; the electric-field-induced second-harmonic generation at 800 nm.

(2) *What has **NOT** been done in Ref.[30]:* Obtaining the near-field THz emission with transform limited bandwidth ($\sim 0.2 - 12$ THz) and complete phase information; providing the quantitative ultrafast photocurrents; measuring the wavelength-dependent THz emission and photocurrent; theoretically revealing the influence of band topology of the Weyl semimetal to the photocurrent.

***We have studied those aspects mentioned above which have not been done in Ref. [30]. The main advances in our work are shown below.

(1) For studying second-order photocurrents (CPGE and LPGE), *analysis of crystal symmetry via group theory and the detail band topology/structures* are crucial. The former tells the existence of these photocurrents, and the latter determines their associated amplitudes. Ref. [30] only addressed the former issue. The latter issue is much more complicated to investigate. In order to explore this problem, quantitatively extracting the photocurrents and measuring the photon-energy dependence must be required.

In semimetal TaAs, we, for the first time, performed experiments over a broad photon energy range 0.5-1.9 eV and provided exact amplitudes of the ultrafast photocurrents. Both of these aspects were not discussed in LANL’s work (which was not published at the time of our submission). We experimentally demonstrated that a resonance behavior resides near ~ 1.5 eV which yields the most efficient CPGE-induced photocurrent generation within the investigated spectral range.

These new results together with our theoretical calculations enable us to reveal how the peculiar band structures near the Weyl nodes affect the photocurrent generation.

(2) We have theoretically calculated the generated CPGE photocurrent as a function of incident light wavelength upon excitation of high photon energies in TaAs for the first time. Our work reveals that the low energy excitations – the chiral fermions in the Weyl bands near E_f - surprisingly dominate the giant photocurrent response even when the photon energy is very high (in the near-infrared regime). This is mainly attributed to the excitation between the anisotropic chiral Weyl band and the excited band accompanied by a large and rapid change in the effective velocity of the charged quasiparticles, and hence by a giant coherent THz radiation. Accuracy of our calculations is confirmed by the experiments: a) The calculated photocurrent amplitude consists well with the experimental data. b) We predict that as the photon energy is tuned away from ~ 1.5 eV, the photocurrent will decay fast, precisely as evidenced by our experiments.

(3) We demonstrate that it is extremely powerful and convenient to control the polarization of **ultra-broadband** THz wave for the Weyl semimetal TaAs. We have directly obtained the correct bandwidth and phase of near-field THz wave (or ultrafast photocurrent), which are vital for extracting the ultrafast photocurrents and future THz applications. All these features were not disclosed in previous work [30]. We discover that this type of THz emitter has many superior advantages, such as large electric field comparable to the commercial THz nonlinear crystals, simple and arbitrary polarization control with ultra-broadband width and high dynamical range. These findings for the first time provide an exceptional example of the Weyl semimetals being used in a practical application, and also open a new route to realize chiral photon sources using quantum materials.

From my point of view, the main advance of this paper (compared to Ref. [30]) is its quantitative computation of the injection current based on the real band structures of TaAs. This quantitative computation is very complicated due to the high photon energy used. Does the schematic in Fig. 7a reflect the real situation? Why is there no excitation from lower bands (occupied) to the Weyl bands above the Fermi level (unoccupied)? Moreover, the authors argued that the mechanism is different from the previous studies with lower photon energy that connects only Weyl bands. I don't understand such an argument. Both are due to the combination of optical selection and the asymmetric bands. The band velocity difference is always there if only considering a one-photon excitation event. Even for a perfect Dirac cone as in graphene, if we only consider one excitation event by one photon, the electron has opposite velocities before and after being excited. The total current may be canceled out by considering the space-time symmetry, which is reflected in the symmetry and angular momentum index of the bands.

We agree with the referee that the calculations for high photon energy are very complicate. The schematic in Fig. 7a is just a simplified picture. It emphasizes our observation arising from the interband optical transitions between the Weyl cones and

the massive bulk bands. The massive bands can be the high-lying bands far above E_f or the low-lying bands far below E_f (we included this information in the revised Fig.7a), as also mentioned by the referee. We derive the expression for CPGE in the situation where optical transition occurs from a linear Weyl band to a massive band. The calculation would be analogous for transitions from a massive band to a linear Weyl band. This is because compared to the electrons, the holes have opposite helicity and charge, which result in the final expression having the same sign.

As describe in the manuscript, in order to avoid tedious calculations but still catch the main physics, we have simplified the massive bulk bands by using the near-flat band assumption (E independent on \mathbf{k}) in our theoretical model. Clearly, treatment in this way might cause the calculated photocurrent not very accurate, but correct order of the current density can be obtained, as well confirmed by our experiments.

Unlike the previous works [C.-K. Chan et al., PRB 95, 041104(2017);Q. Ma et al., Nat. Phys. 13, 842(2017)], our theoretical model does not rely on the Pauli blockade. Here, the key factors determining the injection photocurrent density $J_i(\omega)$ at high photon energy ($\hbar\omega$) are the net effect of the tilted anistropic Weyl cone, its chirality, and the large band velocity difference after excitation. In specific, our model derives $J_i(\omega)$ as [see the main text and Supplemental Materials for details]

$$J_i(\omega, \vec{k}_p, \vec{\epsilon}) = \frac{-eI}{\hbar\omega} \frac{\sum_l \tau_l a_l \chi_l N_{(l)j}^i L^j}{\sum_l a_l D_{(l)}^{ij} \epsilon_i \epsilon_j^*}$$

The above equation has the parameter of chirality χ_l . It also has information about the Weyl cones and band velocities, which are imbedded in the tensors $N_{(l)j}^i$ and $D_{(l)}^{ij}$. If we take the sum over a set of cones with different chiralities, the symmetric components of $N_{(l)j}^i$ cancel due to the tetragonal symmetry of the crystal and the only possible non-canceling contribution is from antisymmetric x-y component $N_{(l)y}^x - N_{(l)x}^y$, which is non-zero only if the tilt Hamiltonian H_t is non-zero, the untilted part of the Hamiltonian H_w is anisotropic, and the tilt is not aligned with principal axes of the untilted part.

In the revised manuscript and Supplemental Material, we include more detail information about our calculations.

2. Another possible novel aspect of this paper is the excitation photon energy dependence, as mentioned in their abstract “The photocurrent generation is maximized at near-IR frequency range close to 1.5 eV”. However, this was not given sufficient discussions in the paper. Since the authors talk about all CPGE (injection current), LPGE (shift current) and thermal current generation, which photocurrent is maximized at 1.5 eV? Are they all enhanced at the same excitation photon energy? PRB 98, 165113 (2018) reported a resonant enhancement of second harmonic

response at ~ 1.5 eV (with ~ 0.7 eV as the principle excitation). Do the authors see similar photon energy dependence for the shift current?

We thank the referee for pointing out this aspect. In the revised manuscript, we added a short discussion about the SHG. We also provided the photon-energy dependent photocurrent density due to the LPGE and photo-thermal effect. Indeed, the photon-energy dependent current $J_x(\omega)$ due to CPGE maximizes at ~ 1.5 eV, which is approximately twice the fundamental resonance energy at which the second harmonic generation peaks [S. Patankar et al., PRB, 98, 165113 (2018)]. The photon-energy dependent giant SHG in Ref. [S. Patankar et al., PRB, 98, 165113 (2018)] is tentatively explained by the nonlinear shift current response due to the skewness of the polarization distribution function in the ground state. Based on those SHG studies, it is still unclear whether their observed behavior is related to the chiral Weyl fermions or not. Although investigations in our work elucidate that the chiral Weyl nodes are responsible for the giant CPGE-induced photocurrent in the near-infrared regime, we are unable to access the relevance to the SHG to our observations since our theoretical model only deals with the CPGE and cannot apply to the shift current.

Fig. 1 (a) Photon-energy ($\hbar\omega$) dependent sheet photocurrent densities J_{yz} and J_x (LPGE). (b) Light absorption coefficient derived from the theoretical dielectric constants [M. Dadsetani, and A. Ebrahimian, J. Electronic Mater., 45, 5867 (2016)].

We also measured the photon-energy dependent LPGE and thermal related current densities (see Fig. 1(a) above). Compared to the CPGE, LPGE-related current shows quite different behavior. Instead of showing resonance behavior around 1.5 eV, $J_x(\omega)$ arising from the LPGE seems to show a plateau-like behavior in the near-infrared regime. This result also differs from the frequency-dependent SHG signals. Because at this stage we do not have an appropriate theoretical model to explain the data, we have to leave it for the future studies. On the other hand, $J_{yz}(\omega)$ continuously increases with $\hbar\omega$. This behavior is very similar with that of the light absorption coefficient $\alpha(\omega)$ (see Fig. 1(b)). Such observation further confirms its thermal origin.

Summary of changes:

(1) We modified our manuscript to address clearly the new findings in this work and clarify the difference between our work and previous works, i.e. discovering the tunable chiral (or helical) THz source with ultra-broadband width, revealing the chiral Weyl fermions indispensable for the predominant CPGE-induced photocurrent via photon-energy dependent investigations over the wide mid- and near-infrared spectral regimes.

Abstract: “...Here, we discover strong coherent terahertz emission from Weyl semimetal TaAs, which is demonstrated as a unique broadband source of the chiral terahertz wave. The polarization control of the THz emission is achieved by tuning photoexcitation of the colossal ultrafast photocurrents via the photogalvanic effect. In the near-infrared regime, the photon-energy dependent nonthermal current due to the predominant circular photovoltaic effect can be attributed to the radical change of the band velocities when the chiral Weyl fermions are excited during selective optical transitions between the tilted anisotropic Weyl cones and the massive bulk bands. Our findings provide an entirely new design concept for creating chiral photon sources using quantum materials and open up new opportunities for developing ultrafast opto-electronics using Weyl physics...”

At the end of 2nd paragraph of page 3 and 1st paragraph of page 4: “...Therefore the investigation of photocurrents in WSMs has raised enormous interest both theoretically and experimentally [20-28]. For mid-infrared light, Refs [20,21,25] show the existence of a dominant helicity-dependent DC photocurrent due to the circular photogalvanic effect (CPGE) in the WSM TaAs. In contrast, with linearly polarized light, a giant linear photogalvanic effect (LPGE) (or shift current) was observed in [26]. At the near-infrared regime, photocurrent measurements [27] suggest the existence of CPGE in TaAs. The THz wave emission from WSM TaAs using ~1.5 eV photon was also reported recently [29], which is interpreted to arise predominantly from the CPGE-induced photocurrents. Similar THz emission was also observed in the ferroelectric semiconductors [30,31], where the LPGE across the bandgap in a wide photon energy range has been studied. However, up to now for WSMs upon excitation of high photon energies with an order of ~1 eV, it is still quite elusive whether the Weyl physics plays an essential role in the giant nonlinear optical responses including the second harmonic generation [32,33] and the photocurrents. Although analysis of the crystal symmetry can reveal all the possible components of the photocurrent [26,28,29], it fails to provide information about the magnitude or photon-energy dependence of the current. Therefore, a theoretical study of the underlying mechanism along with an experimental investigation across the entire near- and mid-infrared ranges are mandatory.”

In the 2nd paragraph of page 4: “In this paper, we reveal the generation of chiral (or helical) broadband THz waves in the WSM TaAs. We find that the polarization of these THz waves can be easily manipulated without incorporating any THz waveplate.

Such polarization control arises from the colossal ultrafast photocurrents whose direction and magnitude depend on the polarization (circularly or linearly polarized) of the femtosecond optical pulses. For the first time, the photon-energy dependent ultrafast photocurrents in TaAs have been quantitatively determined at the near- and mid-infrared light frequencies. In addition, a careful theoretical treatment suggests that the chiral Weyl fermions indeed play a crucial role in the generation of the ultrafast photocurrents due to the dominant CPGE at high photon energies.”

In the conclusions: “The demonstrated generation of chiral ultrafast photocurrents in WSM TaAs offer unique opportunities for novel THz emission with polarization control. The theory underlying the CPGE-induced current is insightful, predicting the photon-energy dependence and demonstrating the essential role of chiral Weyl fermions, in spite of the transition involving a massive band, where there is no clear notion of chirality. In terms of the applications, the simplicity of polarization control of the ultra-broadband THz wave is extremely powerful and useful. Other advantages of the WSM THz emitter include the low cost in sample preparation and the high THz emission efficiency. We further believe that our observation will benefit the study of other novel phenomena led by the Weyl physics, such as the quantized CPGE [24], and the Weyl-orbit effect[45].”

(2) In 3rd paragraph of page 15, we added short discussions about the SHG experiments, “We note that $J_x(\omega)$ here maximizes at $\hbar\omega \sim 1.5$ eV, which is approximately twice the fundamental resonance frequency at which the second harmonic generation peaks [33]. The frequency-dependent giant SHG in Ref.[33] is tentatively explained by the shift current response (LPGE) due to the skewness of the polarization distribution function in the ground state. Based on the SHG studies, it is still unclear whether their observations are related to the chiral Weyl fermions. Moreover, since our theoretical model only deals with the CPGE and cannot apply to the shift current, we are also unable to access the relevance of the SHG to our results.”

(3) In 2nd paragraph of page 15 and 1st paragraph of page 16, we include the calculations of the Glass coefficient for CPGE- and LPGE-induced photocurrents, respectively. **(a)** “The observed photocurrent response is very strong and can be verified by the Glass coefficient (G), as done in Ref.[26]. For instance, at 800 nm we can obtain G to be ~ 10 cm/V and $\sim 4 \times 10^9$ cm/V corresponding to $\max(J_x)$ and \bar{J}_x , respectively. The former is enormously huge, and larger than all the known values [26] by a factor of $\sim 10^9$ - 10^{10} . Such giant value directly reflects an ultrafast current pulse with ps timescale. Indeed, the latter, corresponding to the DC photocurrent, is also above the average G value for a large collection of materials in the near-infrared region [26].” **(b)** “...Similarly, we can get the Glass coefficient (G) to be ~ 3 cm/V and $\sim 10^9$ cm/V corresponding to the maximum ultrafast and equivalent DC LPGE-induced currents at 800 nm. These values are also colossal.”

(4) We changed Fig. 7, which is only for the CPGE happening between the linear Weyl bands and the massive bulk bands. In Fig. 7a, we have included the excitation from the low-lying massive band to the linear Weyl band.

(5) We added Fig. 8, which focuses on the LPGE-induced current. We have included the photon-energy dependent photocurrents arising from the LPGE and thermal effect.

(6) In 1st and 2nd paragraphs of page 17, we added discussions about the frequency-dependent current due to LPGE and thermal effect, respectively.

(a) *“Moreover, we measured the photon-energy dependent LPGE response (see Fig.8b), which, however, behaves quite differently with that of the CPGE-induced current. In specific, instead of showing resonance behavior around 1.5 eV, $J_x(\omega)$ arising from the LPGE seems to show a plateau-like behavior in the near-infrared regime. This result also differs from the frequency-dependent SHG signals [33]. Furthermore, it can hardly be explained by the THz emission observed in the ferroelectric semiconductors with well-defined energy gaps [30,31] because no strong absorption feature in the mid- and near-infrared regimes was reported in the semimetal TaAs (see $\hbar\omega$ -dependent absorption coefficient in the Supplemental Material). Theoretically, it is quite challenging to calculate accurately the frequency-dependent LPGE. At this stage we do not have an appropriate model to explain the data, and thus leave them for future studies. Nevertheless, our experiments demonstrate that the LPGE-induced ultrafast photocurrent can also be significant, and provide an additional control degree of freedom for the broadband THz pulses using linearly polarized light.”* **(b)** *“...Fig.8b shows that $J_{yz}(\omega)$ continuously increases with $\hbar\omega$. This behavior is very similar with that of the light absorption coefficient $\alpha(\omega)$ (see Supplemental Material), and further confirms its thermal origin...”*

(7) Starting from the 1st paragraph on page 13 and ending in the 2nd paragraph on page 14, we have presented more clearly our theoretical model, i.e. detail explanations of the physical parameters inside Eqs.(4) and (5). We also provided the expressions for the tensors N and D in the Supplemental Material.

(8) In 1st paragraph of page 11, we added a brief description of our theoretical calculations.

(9) In the Supplemental Material, we have added the illustration figures (Figs. S3 and S4) for the Weyl cone with different chiralities and the tilted anisotropic Weyl cone considered during our calculations.

(10) In the Supplemental Material, we added the absorption coefficient as a function of the photon-energy $\hbar\omega$, extracted from Ref.[M. Dadsetani and A. J. Ebrahimiyan, J. Electronic Mater., 45, 5867 (2016).]

(11) We have updated some references and included some new ones in the revised manuscript.

[26] Osterhoudt, G. B. et al. Colossal mid-infrared bulk photovoltaic effect in a type-I Weyl semimetal. *Nat. Mater.* 18, 471 (2019).

[28] Ma, J., Gu, Q., Liu, Y., Lai, J., Peng, Y., Zhuo, X., Liu, Z., Chen, J., Feng, J., Sun, D., Nonlinear Photoresponse of Type-II Weyl Semimetals, *Nat. Mater.* 18, 476 (2019).

[29] Sirica, N. et al. Tracking ultrafast photocurrents in the Weyl semimetal TaAs using THz emission spectroscopy. *Phys. Rev. Lett.* 122, 197401 (2019).

[30] Sotome, M., Nakamura, M., Fujioka, J., Ogino, M., Kaneko, Y., Morimoto, T., Zhang, Y., Kawasaki, M., Nagaosa, N., Tokura, Y. and Ogawa, N. Spectral dynamics of shift current in ferroelectric semiconductor SbSI. *Proc. Natl. Acad. Sci.* 116, 1929 (2019).

[31] Sotome, M., Nakamura, M., Fujioka, J., Ogino, M., Kaneko, Y., Morimoto, T., Zhang, Y., Kawasaki, M., Nagaosa, N., Tokura, Y. and Ogawa, N. Ultrafast spectroscopy of shift-current in ferroelectric semiconductor Sn₂P₂S₆. *Appl. Phys. Lett.* 114, 151101 (2019).

[33] Patankar, S., Wu, L., Lu, B., Rai, M., Tran, J. D., Morimoto, T., Parker, D. E., Grushin, A. G., Nair, N. L., Analytis, J. G., Moore, J. E., Orenstein, J. and Torchinsky, D. H. Resonance-enhanced optical nonlinearity in the Weyl semimetal TaAs. *Phys. Rev. B* 98, 165113 (2018).

Reviewers' Comments:

Reviewer #1:

Remarks to the Author:

the authors have done a very good job responding to referees. There are only two minor issues...

In the response, they claim a novelty is their use of symmetry to separate thermal from non-thermal. In fairness, this was done by the Burch group in the NAT. Materials paper. However, it is true for THz emission this has not been as carefully done as here. Related to this, the authors' finding for the Glass coefficient being significantly smaller than the work of Burch et al, makes sense given the much higher energies employed here. This should be noted in the manuscript...

Also, the authors claimed the LPGE does not have a model. This is not correct, it has been calculated many times. Also given it doesn't have a dependence on τ it is usually more reliable. What is really different between CPGE and LPGE, other than τ , is the dependence on differences in the dependence on berry connection (LPGE) vs berry curvature (CPGE). It's not inconceivable that for transitions measured here the CPGE can be larger.

Reviewer #2:

Remarks to the Author:

I would like to thank the authors for the detailed reply. I think that the authors have addressed all my questions. I recommend the paper for publication.

Reviewer #3:

Remarks to the Author:

I believe my concerns have been successfully addressed by the authors and therefore I support its publication in Nature Communications now.

Response to Reviewer #1:

We thank the referee for giving high remark to our response, and providing us further suggestions to improve our manuscript. Below, we list the referees' comments in blue italics and our response in normal font. At the end of this document, we have provided a summary of changes.

the authors have done a very good job responding to referees. There are only two minor issues...

In the response, they claim a novelty is their use of symmetry to separate thermal from non-thermal. In fairness, this was done by the Burch group in the NAt. Materials paper. However, it is true for Thz emission this has not been as carefully done as here. Related to this, the authors finding for the Glass coefficient being significantly smaller than the work of Burch et al, make sense given the much higher energies employed here. This should be noted in the manuscript...

According to the referee's comments and suggestions, we have revised our manuscript: (1) In the introduction (2nd paragraph), we have clearly stated that in the work by Burch group, symmetry analysis enables to extract the *non-thermal* photocurrent. (2) As suggested by the referee, we have added a note to articulate and explain the observed smaller Glass coefficient.

Also, the authors claimed the LPGE does not have a model. This is not correct, it has been calculated many times. Also given it doesn't have a dependence on tau is usually more reliable. What is really different between CPGE and LPGE, other than tau, is the dependence on differences in the dependence on berry connection (LPGE) vs berry curvature (CPGE). Its not inconceivable that for transitions measured here the CPGE can be larger.

(1) The referee might misunderstand our previous response about the LPGE calculations. We actually did not claim that “the LPGE does not have a model”. We only stated, ‘we do not have a model’. In order to get rid of such confusion, we rewrote the corresponding sentence in the maintext (2nd paragraph of page 17). (2) We also have adapted the referee's suggestions which explain why CPGE is larger in our experiments, and incorporated them into our text.

Summary of changes:

Changes according to 1st referee's suggestions/comments

(1) In 2nd paragraph of introduction, we have clearly stated that symmetry analysis helped to extract the non-thermal photocurrent in the work by Burch group (Ref.[26]).

“... analysis of the crystal symmetry can reveal all the possible components of the non-thermal photocurrent [26,28,29]...”

(2) At the bottom of last paragraph on page 15, we have added the explanation of the observed Glass coefficient, as presented by the referee, *“We note that our calculated Glass coefficient corresponding to the equivalent DC photocurrent is still significantly smaller than that associated with the shift current in Ref.[26]. Nevertheless, our obtained G value is reasonable given the much high photon energies employed here.”*

(3) In 2nd paragraph of page 17, we have rewritten the sentence about describing the calculation model of LPGE, which becomes, *“Theoretically, it is quite challenging to calculate accurately the frequency-dependent LPGE at high photon energies, and we thus leave it for future studies.”*

(4) In 2nd paragraph of page 17, we have incorporated the referee’s suggestions which explain why CPGE is larger in our experiments, *“We note that the difference between LPGE and CPGE might arise from two factors: (1) Independence on the relaxation time τ for LPGE is usually more reliable. (2) LPGE depends on the berry connection while CPGE relies on the berry curvature. As a results, it is conceivable that the CPGE-induced photocurrent can be larger in our measurements.”*